# Stroke induces disease-specific myeloid cells in the brain parenchyma and pia

Carolin Beuker[1,8], David Schafflick[1,8], Jan-Kolja Strecker[1,8], Michael Heming [1], Xiaolin Li[1], Jolien Wolbert[1], Antje Schmidt-Pogoda[1], Christian Thomas [2], Tanja Kuhlmann [2], Irene Aranda-Pardos[3], Noelia A-Gonzalez[3], Praveen Ashok Kumar[4], Yves Werner[4], Ertugrul Kilic[5], Dirk M. Hermann[6], Heinz Wiendl [1], Ralf Stumm [4], Gerd Meyer zu Hörste [1,9✉] & Jens Minnerup [1,7,9✉]

Inflammation triggers secondary brain damage after stroke. The meninges and other CNS border compartments serve as invasion sites for leukocyte influx into the brain thus promoting tissue damage after stroke. However, the post-ischemic immune response of border compartments compared to brain parenchyma remains poorly characterized. Here, we deeply characterize tissue-resident leukocytes in meninges and brain parenchyma and discover that leukocytes respond differently to stroke depending on their site of residence. We thereby discover a unique phenotype of myeloid cells exclusive to the brain after stroke. These stroke-associated myeloid cells partially resemble neurodegenerative disease-associated microglia. They are mainly of resident microglial origin, partially conserved in humans and exhibit a lipid-phagocytosing phenotype. Blocking markers specific for these cells partially ameliorates stroke outcome thus providing a potential therapeutic target. The injury-response of myeloid cells in the CNS is thus compartmentalized, adjusted to the type of injury and may represent a therapeutic target.

[1] Department of Neurology with Institute of Translational Neurology, Medical Faculty, University Hospital, Münster, Germany. [2] Institute of Neuropathology, University of Münster, Münster, Germany. [3] Institute of Immunology, Westfälische Wilhelms-University, Münster, Germany. [4] Institute of Pharmacology and Toxicology, Jena University Hospital, Jena, Germany. [5] Istanbul Medipol University Regenerative and Restorative Medical Research Center, Istanbul, Turkey. [6] Department of Neurology, University Hospital Essen, Essen, Germany. [7] Interdisciplinary Center for Clinical Research (IZKF), Münster, Germany. [8] These authors contributed equally: Carolin Beuker, David Schafflick, Jan-Kolja Strecker. [9] These authors jointly supervised this work: Gerd Meyer zu Hörste, and Jens Minnerup. ✉email: gerd.meyerzuhoerste@ukmuenster.de; minnerup@uni-muenster.de

The meningeal layers that envelop the brain parenchyma and the choroid plexus (CP) that produces the cerebrospinal fluid (CSF) have long been known to provide trophic support and mechanical protection to the central nervous system (CNS)[1]. However, recently unexpected immune-related functions of these CNS border compartments were identified[2,3]. Even in homeostasis, the meninges induce a border-associated cellular phenotype in myeloid lineage cells[4,5]. Similarly, tissue-resident lymphocytes also exhibit a location-specific composition and phenotype in each individual compartment[4]. For example, we and others previously identified that the dura contains a large proportion of B cells across a broad developmental spectrum[6,7] and meningeal lymphocytes form local immune hubs[8]. Immunity in border compartments also responds to diseases, as shown for meningeal leukocytes in experimental neuroinflammation[4,6,9] and in traumatic brain injury[9]. And meningeal lymphatic vessels in fact partially control neuroinflammation[5,10] and neurodegeneration[11].

However, the understanding of how the different CNS-associated border compartments quantitatively and qualitatively respond to different types of injury is limited. Previous studies found that experimental neuroinflammation induces a specific phenotype of 'inflammation-associated' myeloid cells in the CNS parenchyma that is partially shared by myeloid cells in border compartments[5,12]. A myeloid cell population with similar naming ('disease-associated microglia'), but a different phenotype was identified in Alzheimer disease models and in corresponding human tissue samples. Such 'degeneration-associated' microglia were controlled by the APOE-TREM2 pathway and are considered to modify neurodegeneration and thus may represent a future therapeutic target in neurodegenerative diseases[13,14]. But whether myeloid cell responses induced by other types of CNS tissue injury are identical or distinct, and whether this is shared in the meninges is unknown.

Stroke, a leading cause of death and disability worldwide, induces a robust inflammatory response of multiple leukocyte lineages in the CNS parenchyma[15] that contributes to secondary tissue injury and can impede recovery in stroke[16,17]. However, the relevance of CNS-associated border compartments in stroke-induced neuroinflammation is poorly understood. In a previous study, the choroid plexus was suggested as an invasion site for T cells after stroke[18]. It was also shown that gut microbiota control how leptomeningeal IL-17+ γδ-T cells and intestinal T cells traffic to the leptomeninges in stroke[19]. But whether stroke-associated cellular immunity in border compartments is overall similar or distinct from the CNS parenchyma is unknown.

In this work, by providing a comprehensive cellular census of leukocytes in CNS and its border compartments in stroke, we demonstrate that ischemic stroke affects not only CNS tissue-resident but also meningeal cellular immunity. We then discover a specific stroke-associated phenotype of myeloid cells in the pia and CNS parenchyma in rodents and humans exhibiting a lipid-phagocytosing phenotype. These stroke-associated myeloid cells (SAMC) share features with microglia in neurodegenerative diseases and from development and are mostly, but not exclusively, derived from resident microglia. Moreover, we find that blocking SAMC-specific markers partially ameliorates stroke outcome. The injury-response of myeloid cells in the CNS is thus compartmentalized to specific locations and is not stereotypical, but rather individualized and fine-tuned to the type of injury and may represent a therapeutic target.

## Results
### Stroke induces a defined leukocyte response in CNS parenchyma and associated border compartments. Stroke induces an inflammatory response in the CNS parenchyma that is dominated first by myeloid lineage cells and second by lymphocytes[15,20]. Whether the same is true for CNS-associated border compartments was unknown. We therefore induced transient middle cerebral artery occlusion (MCAO) in C57BL/6 mice and then characterized cellular immunity in multiple border compartments and CNS parenchyma. We used two artery occlusion times (30 and 45 min) and two post-ischemia analysis times (24 and 72 h) to model different conditions (Fig. 1A). As expected, infarct volumes were highest at 72 h and increased with longer artery occlusion (Supplementary Fig. 1A). This was reflected by a time-dependent increase of apoptotic neurons that was higher in striatum than in cortex (Supplementary Fig. 1B) as expected in the model. We then used flow cytometry to quantify leukocyte subsets after stroke (Supplementary Fig. 2). We enriched tissue-resident leukocytes (TRLs) and reduced blood-derived leukocytes by intravenous injection of CD45 fluorescent antibody before perfusion and subsequently specifically analyzed CD45iv−CD45+ TRL, as described previously[6,21].

Using flow cytometry, in non-ischemic brain parenchyma, microglia (defined as CD45intCD11b+) were the most abundant cell type, followed by monocytes/macrophages (mono/macros; CD45+CD11b+Ly6-C/G+F4/80+), dendritic cells (DC; CD45+CD11b+CD11c+), T cells (Tc; CD45+CD3+), granulocytes (granulo; CD45+Ly6-C/G+F4/80−), B cells (Bc; CD45+B220+CD11c−CD11b−), and natural killer cells (NK; CD45+NK-1.1+) (Fig. 1B). This composition was in accordance with expectations[15].

At 24 h and 72 h after ischemia, we found a significant increase of DCs, monocytes/macrophages, and granulocytes in the brain parenchyma over time while the proportion of microglia of all leukocytes decreased probably due to the influx of leukocytes. T cells only started to increase from >24 h on, while NK cells remained unchanged. B cells showed a non-significant tendency towards increasing numbers over time as well (Fig. 1B).

We next extended our analysis to border compartments with special focus on the meninges (Fig. 1B). As we and others have previously shown[4,6], tissue-resident leukocytes in the healthy meninges and the other border compartments were distinct from the parenchyma and overall the responses to ischemia were tissue-specific. In all tissues, B cells and T cells were essentially unchanged while monocytes/macrophages showed an increase from 24 h on in the CP (choroid plexus). NK cells were decreased at 72 h in the CP (Fig. 1B) while granulocytes show a temporary increase at 24 h in the pia. In the dura, DCs significantly decreased over time post-ischemia, while NK cells showed a decrease at 24 h. Granulocytes increased in the pia at 24 h (Fig. 1B; Supplementary Figs. 2, 3).

We next used immunohistochemistry to evaluate the spatiotemporal distribution of the infiltrated myeloid cells (F4/80+), granulocytes (7/4+), Tc (CD3+), and Bc (B220+) (Supplementary Fig. 1C). As expected from flow cytometry, all leukocyte subsets numerically increased, with a cortex-to-striatum gradient and myeloid cells and granulocytes were most abundant (Supplementary Fig. 1D). This confirms that stroke induces an initially myeloid lineage-dominated inflammatory response in the CNS parenchyma with a second wave response mediated by T cells.

Furthermore, we aimed to histologically confirm the most apparent changes of leukocyte composition and quantified the number of granulocytes in the combined meninges. Different meningeal layers (pia vs. dura) could not be discerned in histology. We found that the density of meningeal 7/4+ granulocytes increased at 72 h after 45 min of MCAO (Supplementary Fig. 1E). Dura and pia thus exhibit a site-specific leukocyte response to stroke that differs from parenchyma and shows a temporally defined pattern.

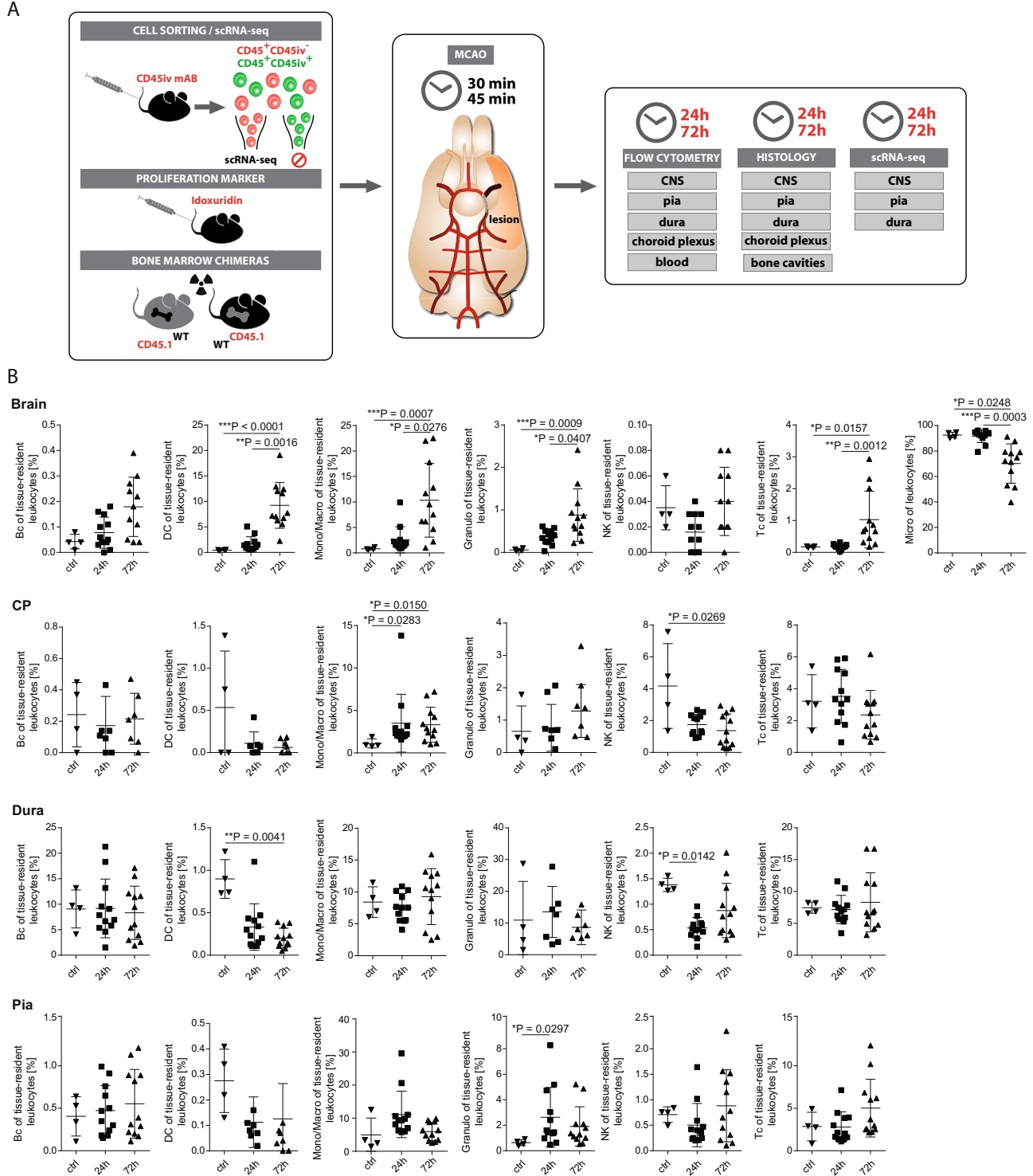

**Fig. 1 Stroke affects leukocytes in a compartment-specific fashion. A** Scheme of the experimental approach. Middle cerebral artery occlusion (MCAO) (or sham-operated, as controls) was induced in wild-type (WT) mice for 30 or 45 min and mice were sacrificed 24 h or 72 h later (post ischemia). Prior to stroke induction, fluorophore-labeled CD45 antibody (mAb) was injected intravenously (iv). After 5 min, mice were intracardially perfused and CD45+-leukocytes negative for the CD45iv antibody (CD45+CD45iv−) then defined as tissue-resident leukocytes were flow-sorted from central nervous system (CNS) parenchyma, pia, dura, choroid plexus (CP) and blood, and analyzed by flow cytometry and single-cell RNA-sequencing (scRNA-seq). Tissue sections were analyzed by immunohistochemistry. **B** Frequencies of Bc (B cells), DC (dendritic cells), Mono/Macro (Monocytes/Macrophages), Granulo (Granulocytes), NK (Natural killer cells), Tc (T cells) and Micro (Microglia) of tissue-resident leukocytes isolated out of the Brain, CP, Dura and Pia from sham-operated mice without stroke (ctrl), from mice 24 h post ischemia (24 h) and from mice 72 h post ischemia (72 h) analyzed by Flow Cytometry. $n = 4$ for ctrl, $n = 12$ for 24 h and 72 h. Data are presented as mean values ± SD. Statistical significance was tested using Kruskal–Wallis test with Dunn's post test. Not significant = not shown, *$P ≤ 0.05$, **$P ≤ 0.01$, ***$P ≤ 0.001$. Source data for B are provided as a Source Data file.

**Single-cell transcriptomics identifies specific stroke-associated myeloid cells in CNS and pia.** We speculated that flow cytometry would miss more subtle compositional changes induced by ischemia over time. We, therefore, performed single-cell RNA-sequencing (scRNA-seq) of CD45iv−CD45+ TRL sorted from the brain parenchyma, pia, and dura at 24 h and 72 h after MCAO compared to healthy control animals. We thereby obtained 32,457 total single-cell transcriptomes from the CNS (18,396 cells), pia (6,650 cells), and dura (7200 cells) with 8050 cells derived from CNS at 24 h and 6,049 cells derived from 72 h (Supplementary Data 1). We jointly clustered data from all tissues at 24/72 h after ischemia together with controls and identified 21 cell clusters (Fig. 2A) that were annotated based on marker genes (Fig. 2B).

Lymphoid lineage clusters separated into Tc (*Cd3g, Trbc2, Cd8a*), NK cell (*Nkg7, Gzma*), Bc (*Cd79a,* Ms4a1), γδ Tc (gdTc; *Cd3g, Trdc*) and innate lymphoid cells type (ILC2; *Gata3, Rora*) (Fig. 2A, B). Myeloid lineage clusters were separated into 15 clusters (Fig. 2A). Four clusters expressed microglia transcripts (micro_1-3, stress_Micro; *Tmem119, Hexb, Cx3cr1, Sparc, P2ry12, Cst3*) and were almost exclusively found in the brain (Fig. 2C; Supplementary Fig. 4A, C). Five clusters expressed macrophage transcripts (CAM_1, CAM_2, SAMC, Macro_1, Macro_2; *Lyz2, Apoe*) of which one cluster expressed a pronounced border-associated phenotype (CAM_1; *Pf4, Lyve1*) while one cluster showed a weaker expression of CNS border-associated transcripts (CAM_2; *Pf4*). The macrophage clusters were mainly present in pia and dura and lower in brain parenchyma (Fig. 2C; Supplementary Fig. 3A, C). One cluster (SAMC; *Cd14, Apoe, Spp1, Lpl*) (Fig. 2A, B) was notably mainly present in brain parenchyma after ischemia and absent in the other compartments (Fig. 2C; Supplementary Fig. 3A, C).

Additionally, we identified clusters expressing genes indicating granulocytes (granulo_1, granulo_2; *Lyz2, Fn1, Hp, S100a8*), type 1 myeloid dendritic cells (mDC1; *Ifitm1, Clec9a*) and type 2 myeloid dendritic cells (mDC2; *Cd209a*), mast cells (Mast; *Tpsb2*), stressed myeloid cells (stress_Myeloid; *Hspa1a, Ubc*), and proliferating cells (prolif_cells; *Hmgb2, Birc5*) (Fig. 2A, B). This composition confirms the known border-associated phenotype of myeloid cells in the meninges compared to the CNS[4,5].

We next systematically compared the leukocyte compositional changes during the different time points post-ischemia and different tissues. The brain parenchyma showed an increase of several myeloid lineages during the first 24 h post-ischemia with stress_micro, SAMC and CAM_1 showing the largest increase while clusters annotated as microglia (Micro_1, Micro_2, Micro_3) decreased (Fig. 2C, D). Notably, the SAMC cluster was absent in non-ischemic samples and thus highly stroke specific. We, therefore, annotated this cluster as "stroke-associated myeloid cells" (SAMC). In a 24 h vs. 72 h post-ischemia comparison, Granulo_1, Macro_2 and CAM_2 clusters increased, while Microglia clusters (Micro_1, Micro_2, Micro_3 and stress_Micro) and CAM_1 decreased (Fig. 2C,D). Interestingly, the CAM_2 cluster increased more strongly from 24 to 72 h post-ischemia than from ctrl to 24 h (Fig. 2C). These results may reflect second wave changes towards more specialized granulocytes and specific Macrophage/CAM populations.

In the pia at 24 h post-ischemia, Granulo_1, mDC1 and Granulo_2 increased while Macro_2 and CAM_2 decreased (Supplementary Fig. 4A, B). In the dura, preferentially the ILC2 cluster increased, while Macro_2 and mDC clusters (mDC_1, mDC_2) decreased (Supplementary Fig. 4C, D). In summary, the leukocyte response to ischemia is highly compartment-specific while a unique myeloid cell population (SAMC) occurs in the parenchyma.

**Stroke-associated myeloid cells express a phagocytosing and lipid-sensing phenotype.** We next aimed to transcriptionally better characterize this stroke-specific cell cluster. Comparing gene expression in the SAMC vs. all other clusters, we found similarity with macrophage clusters (*Apoe, Lyz2*) and overlap with the CAM_1 and Macro_2 clusters (*Cd14*) (Fig. 2B; Supplementary Data 2). Canonical microglia genes (*Tmem119, Hexb, Cx3cr1, P2ry12, Cst3, Sparc*) and other myeloid subset markers showed lower expression but were still detectable (Fig. 2B, Supplementary Data 2). This can be interpreted as indicating a mixed macrophage and microglial phenotype of SAMC.

Searching for unique differentially expressed genes in the SAMC cluster, we identified a high expression of *Spp1* (encoding osteopontin), *Fabp5* (encoding fatty acid-binding protein 5), *Gpnmb* (encoding glycoprotein nmb), and *Lpl* (encoding lipoprotein lipase). We also identified the genes *Mmp12* (encoding matrix Metallopeptidase 12), *Csf1* (encoding M-CSF), and *Adam8* (encoding a disintegrin and metalloproteinase domain-containing protein 8) to be preferentially expressed in SAMC (Fig. 3A; Supplementary Data 2).

The top 10 differential expressed genes (*Spp1, Fabp5, Gpnmb, Ctsb, Ctsl, Lgals3, Lpl, Fth1, Cd63, Ctsd*) reliably identified the SAMC cluster (Fig. 3B) even though co-expression of *Lpl* and *Spp1* (Fig. 3C), as well as Fabp5 and Gpnmb (Fig. 3D), sufficed to identify the core SAMC cluster. Notably, the top 4 differentially expressed genes (*Spp1, Fabp5, Gpnmb*, and *Lpl*) are all associated with lipid metabolism and phagocytosis of myelin.

At 72 h post-ischemia, SAMC underwent a slight expressional change characterized by upregulation of classical macrophage-associated genes *Apoe* and *Lyz2* as well as *Fabp5* and *Gpnmb* (Supplementary Fig. 4E; Supplementary Data 3). These genes were thus induced at later time points (>24 h post-ischemia)[22,23].

Transcription factor (TF) protein-protein interaction (PPI) analysis suggested a potential relevance of *ATF2* in the SAMC cluster (see TF PPI tab in Supplementary Data 4). This TF regulates inflammation-induced transcription in macrophages and is expressed by inflammatory macrophages in adipose tissue and in LPS activated microglia[24–26]. Pathway analysis (by KEGG, WikiPathways, Panther) preferentially identified cholesterol metabolism, lysosome- and proteasome-associated transcripts (see Enrichr tab in Supplementary Data 4). Therefore, we speculated that SAMC may contribute to the clearance of damaged myelin from the post-ischemic brain[22,23,27].

**Stroke-associated myeloid cells partially resemble developing and neurodegeneration-associated microglia.** We next systematically analyzed how SAMC were related to previously described myeloid cell phenotypes. Therefore, we compared the differentially expressed genes of the SAMC population to previously described gene signatures identified in macrophages or microglia from murine atherosclerosis[28], amyotrophic lateral sclerosis (ALS)[13], multiple sclerosis (MS)[14], Alzheimer's disease (AD), and experimental autoimmune encephalomyelitis (EAE)[5] but none of these signatures showed a specific or high overlap with the SAMC population (Supplementary Fig. 5A).

Compared with (neuro-)degeneration-associated microglia (DAM)[13,14], we identified a partial, but incomplete transcriptional overlap with the SAMC transcriptome (Supplementary Fig. 5B). In fact, some of the genes (Supplementary Data 5) were shared between the SAMC cluster and DAM (e.g. *Lpl, Spp1, Fabp5, Apoe, Gpnmb, Csf1, Cd9*), some were specific to the SAMC cluster (e.g. *Lgals3, Cd14, Mmp12*), and some were specific to DAM (e.g. *Cst7, Clec7a, Itgax, Axl*) (Supplementary Data 5).

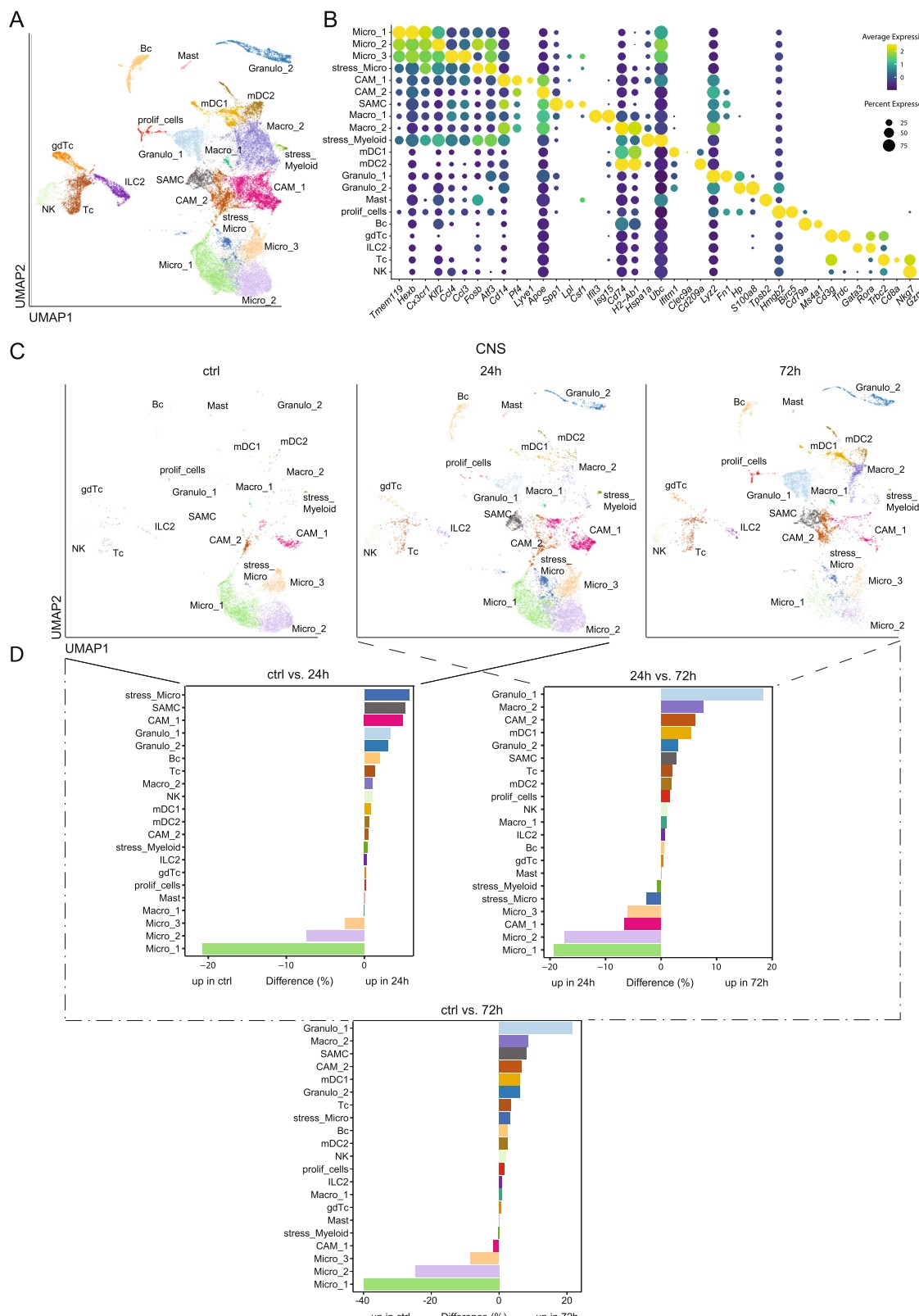

When compared with inflammation-associated macrophages (IAM), the top IAM (e.g. *C1qb, Cst3, Trem2*) genes were not highly expressed by the SAMC (Supplementary Fig. 5C) and SAMC-specific genes (e.g. *Spp1, Fabp5, Gpnmb, Lgals3, Lpl, Apoe*) were not expressed in these macrophages. Thus these results indicate that SAMC have a unique phenotype and therefore seem to be fine-tuned to the type of injury and only partially resemble DAM or IAM.

We next annotated SAMC using data from the mouse cell atlas (MCA)[29]. The SAMC cluster's transcriptomes were mostly annotated as microglia/macrophages, but also consistently as embryonic or developing microglia/macrophages (Fig. 3E). In fact, we found a high overlap of transcripts between previously described early postnatal proliferative-region-associated microglia as well as axon tract-associated microglia (e.g. *Spp1, Fabp5,*

**Fig. 2 Single-cell RNA-sequencing identifies a unique stroke-associated myeloid cell (SAMC) population. A** Merged Uniform Manifold Approximation and Projection (UMAP) plot representing 21 color-coded cell clusters identified in the combined single-cell transcriptomes obtained from brain, dura and pia of sham (ctrl) and MCAO mice (24 h and 72 h post stroke). Cluster names were manually assigned. **B** Dot plot of selected marker genes characterizing the clusters shown in **A**. **C** UMAP plots of single-cell transcriptomes obtained from brain of sham (ctrl) and MCAO mice, 24 h (24 h) and 72 h (72 h) post stroke. **D** Bi-directional histograms depicting differences in relative cluster abundance between MCAO (24 h and 72 h post ischemia) and sham (ctrl) samples as shown in **C**. Differences are calculated as: percentage of all cells in the respective stroke sample minus percentage of all cells in the control sample. Cluster annotations and cluster coloring are consistent between all panels. Abbreviation key: Bc B cell, CAM central nervous system-associated macrophages, CNS central nervous system, gdTc γδ T cells, granulo granulocytes, ILC2 innate lymphoid cells type 2; Macro macrophage, Mast mast cells, mDC myeloid dendritic cell, Micro microglia, NK natural killer cells, proli_cells proliferating cells, SAMC stroke-associated myeloid cells, stress_Micro stressed Microglia; stress_Myeloid, stressed Myeloid cells; Tc, T cells. Source data for D are provided as a Source Data file.

*Gpnmb, Lpl, Lgals3, Apoe, Lilrb4a, Csf1, Ctsb*) (Fig. 3F, G; Supplementary Fig. 5D; Supplementary Data 5). Axon tract-associated microglia are thought to regulate the growth and fasciculation of axons and synapses while showing high lysosomal and phagocytic activity[30]. These data suggest that SAMC could be specific, activated myeloid cells with high lysosomal and phagocytic activity clearing lipid debris similar to microglia found during embryonic development.

**Localization and temporal dynamics of stroke-associated myeloid cells (SAMC).** We next aimed to localize and confirm the SAMC population on protein level. We selected the top SAMC markers and stained coronal brain sections of mice after MCAO for Osteopontin/*Spp1*, LPL/*Lpl*, M-CSF/*Csf1*, and ADAM8/*Adam8* together with pan-myeloid F4/80. We found that Osteopontin/*Spp1* was expressed widely across F4/80-positive cells (Fig. 4A). The markers LPL, M-CSF, and ADAM8 were also consistently expressed by F4/80$^+$ cells within the infarction core (Fig. 4D). In a spatiotemporal analysis, Osteopontin-expressing cells were present across the entire area supplied by the MCA including striatum and cortex and increased within 72 h after MCAO (Fig. 4B, C). Also, the density of LPL$^+$, M-CSF$^+$, and ADAM8$^+$ cells increased and spread from the infarction core to the cerebral cortex (Fig. 4D), confirming expression of the top SAMC markers in ischemic brain parenchyma. We next applied the proliferation marker iododeoxyuridine (IdU) to mice after ischemia and subsequently stained for IdU together with M-CSF. IdU$^+$M-CSF$^+$ cells were frequent within the infarction core indicating proliferation of SAMC (Fig. 4E).

**SAMC exhibit phagocytosing phenotype across species.** As SAMC expressed phagocytosing and lipid-sensing proteins, we speculated that these cells could process damaged myelin. Indeed, M-CSF/FluoroMyelin-co-staining showed a widespread interaction of SAMC with myelin and intracellular vesicular myelin structures in M-CSF$^+$ cells (Fig. 4F). We next labeled oligodendrocytes and neurons together with F4/80. Interestingly, F4/80$^+$, M-CSF$^+$ and LPL$^+$-cells frequently engulfed both oligodendrocytes and whole neurons after stroke (Fig. 4G, H). SAMC may thus serve to phagocytose and remove lipid-rich cell debris after stroke. To experimentally investigate this hypothesis, we first performed histological staining of the SAMC-marker M-CSF with BODIPY, a dye that specifically labels neutral lipids and is commonly used to detect lipid droplets[31]. These identified BODIPY$^+$ lipid droplets were preferentially found in SAMC marker positive cells (M-CSF, Fabp5) within the infarcted area (Supplementary Fig. 6A). Furthermore, we also detected Perilipin-2, a lipid droplet surface protein that is highly expressed by SAMC, frequently expressed among cells with fluorescently labeled myelin indicating phagocytosis of myelin structures (Supplementary Fig. 6A). These findings were conserved in photothrombotic brainstem ischemia in rats (Supplementary

Fig. 6B). To next test whether the SAMC in fact exerted lipid-phagocytosing function, we performed in vitro lipid phagocytosis assays. Compared with non-SAMC CD45$^{high}$Lilrb4$^-$ cells, CD45$^{high}$Lilrb4$^+$ SAMC-specific cells showed increased bodipy uptake indicating lipid phagocytosis (Supplementary Fig. 6D). Moreover, the percentage of CD45$^{high}$Lilrb4$^+$ SAMC cells presenting diffused fluorescence indicating lipid clearance was increased in comparison to non-SAMC cells (Supplementary Fig. 6D). In addition, immunohistochemical staining of human stroke tissue detected SAMC marker positive cells together with BODIPY$^+$ lipid droplets (Supplementary Fig. 6C). Together, these findings validate the phagocytic function of SAMC and confirm that these properties are conserved across species.

**Stroke-associated myeloid cells are replicated in other species, experimental stroke models, and human stroke patients.** We next investigated the SAMC phenotype in other species and stroke models. First, we stained coronal brain sections of mice with the different Balb/C genetic background 24 h after 30 min MCAO for Osteopontin/*Spp1*, LPL/*Lpl*, and M-CSF/*Csf1* together with pan-myeloid F4/80 (Supplementary Fig. 7A). We found all of these markers being consistently expressed by F4/80$^+$ cells within the infarction core. In addition, we induced photothrombotic brainstem ischemia in rats and found the SAMC-markers Osteopontin/*Spp1*, M-CSF/*Csf1*, ADAM8/*Adam8*, and LPL/*Lpl* frequently expressed by IBA1$^+$ cells within the infarcted brainstem parenchyma (Supplementary Fig. 7B). After induction of MCAO in rats, the SAMC-markers Osteopontin/*Spp1*, M-CSF/*Csf1* and MMP12/*Mmp12* were also expressed across Iba1-positive cells that were morphologically classified as microglia or macrophages (Supplementary Fig. 7C). In summary, we replicated the presence of SAMC-like myeloid lineage cells across different models of cerebral ischemia, mouse strains, and species.

We next validated our findings in humans. For this, we used autopsy material of ischemic brain tissue from patients who died within 1–7 days after stroke onset. In line with immunostaining in rodents, we stained human brain sections for Osteopontin, LPL, M-CSF/*Csf1*, ADAM8, and MMP12 in combination with IBA1 to label microglia/macrophages or MAP2 to mark neuronal structures. Osteopontin$^+$ cells were widespread in the infarcted human tissue (Fig. 4I). Additional SAMC-markers LPL, ADAM8, and MMP12 were also frequently expressed by IBA1$^+$ cells that were morphologically classified as microglia or macrophages within the infarcted parenchyma (Fig. 4I). Also in the human samples, SAMC-marker expressing IBA1$^+$ cells showed both ramified and amoeboid shapes indicative of activated microglia and were occasionally seen in cell clusters (Fig. 4I). M-CSF-expressing cells were seen regularly in close vicinity to neuronal MAP2$^+$-structures indicating either neuronal interaction or phagocytosis (Fig. 4J). These results indicate that SAMC are conserved across species and share phenotypic features among rodents and humans.

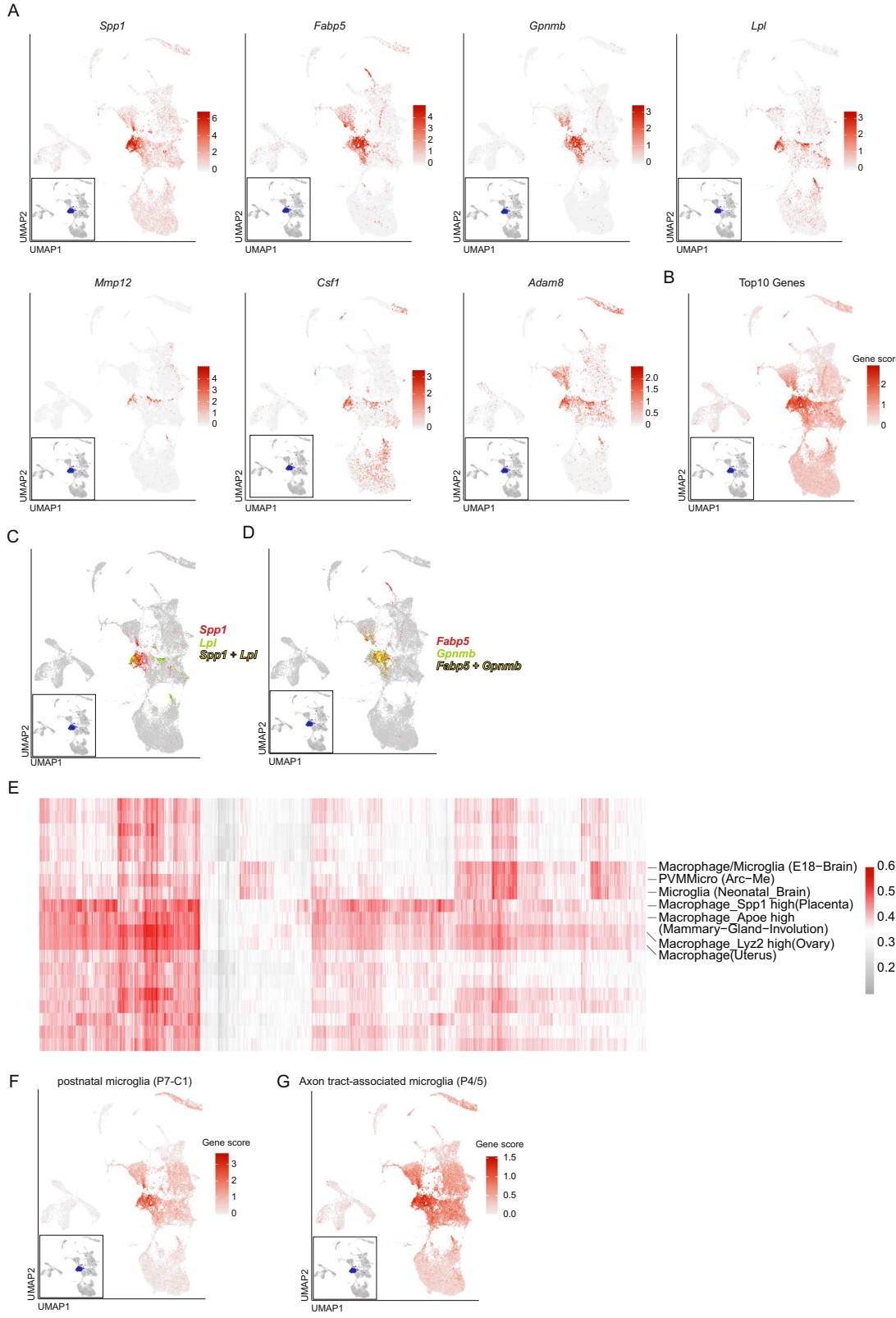

**Stroke-associated myeloid cells are of mixed peripheral and resident microglia origin**. Next, we aimed to address the brain-resident vs. peripheral origin of SAMC in rodents. Therefore, we first generated bone marrow chimeric mice with CD45.2$^+$ microglia and >90% CD45.1$^+$ peripheral macrophages or vice versa (Supplementary Fig. 8A). All mice had >90% chimerism before stroke induction (Supplementary Fig. 8B). We found that M-CSF$^+$ cells in the infarction core were of mixed resident and peripheral origin with slight preponderance of resident myeloid cells (Supplementary Fig. 8C).

In addition, we then utilized a recently described $Cxcr4^{CreERT2}$ x $R26^{CAG-LSL-tdTomato}$ fate-reporting model that almost exclusively

**Fig. 3 The stroke-associated myeloid cell (SAMC) phenotype partially resembles early embryonic microglia and upregulates genes of lipid metabolism. A** Feature Plots of selected marker genes specifically enriched (Methods) in the SAMC cluster; termed stroke-associated myeloid cells in the text. **B** Gene score feature plot of the top 10 genes specifically expressed in the SAMC cluster at 24 h post-ischemia. **C** Dual color feature plot of *Spp1* and *Lpl*. **D** Dual color feature plot of *Fabp5* and *Gpnmb*. **E** All cells assigned to the SAMC cluster were submitted to the mouse cell atlas (MCA) and compared with public single-cell datasets. Each column represents one cell, each row represents one MCA reference cell type. Colors indicate Pearson correlation coefficient between the top MCA cell types and the submitted cells. Labeled are reference datasets with a high Pearson correlation across multiple cells. **F** Gene score feature plot using the Top25 marker genes of postnatal day 7 microglia cells (P7-C1) from[39]. **G** Gene score feature plot using the Top30 marker genes of axon tract-associated microglia cells (P4/5) from[30].

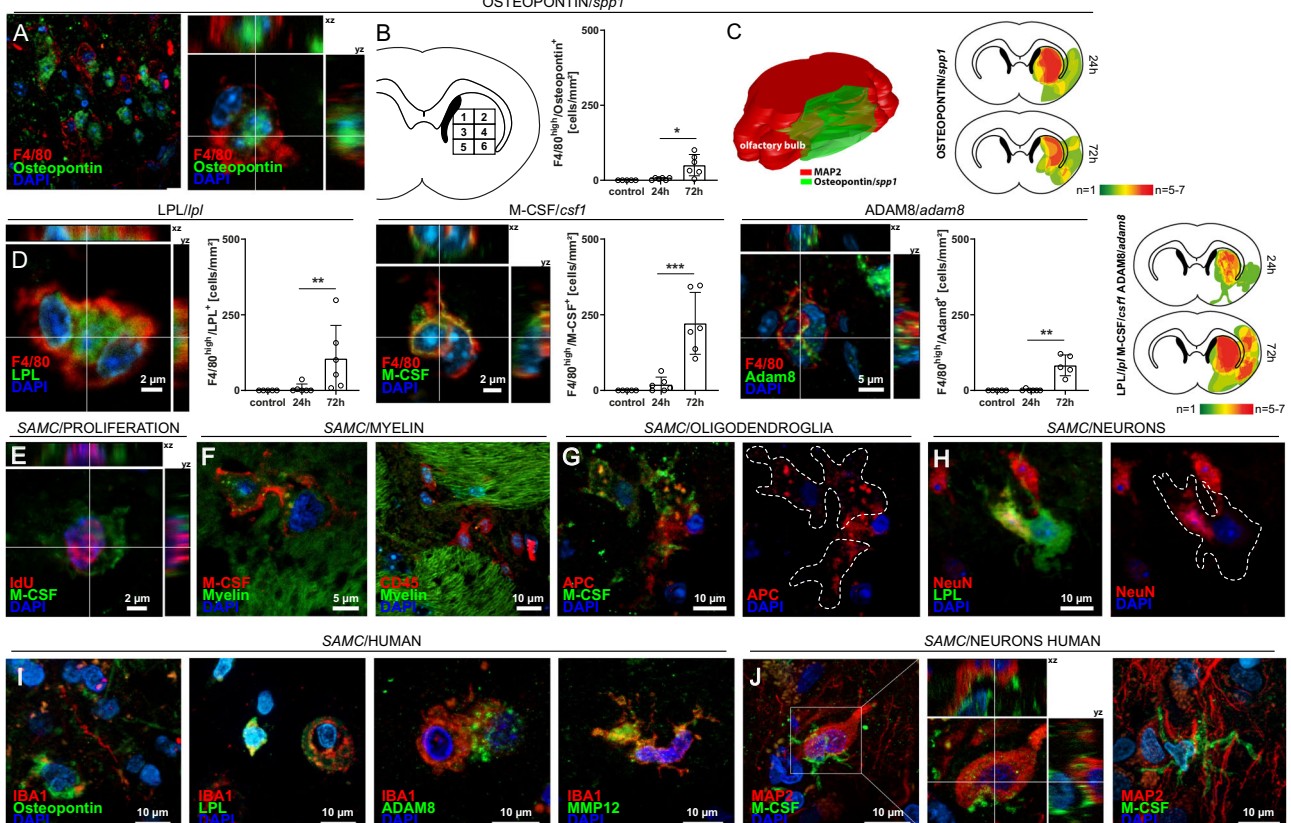

**Fig. 4 Top markers of stroke-associated myeloid cells (SAMCs) are confirmed within infarcted brain parenchyma in rodents and human stroke patients. A** Middle cerebral artery occlusion (MCAO) was induced in C57BL/6 mice for 45 min. Coronal (approx. bregma 0–1 mm) and transversal cryosections were prepared of 24 and 72 hours post-MCAO brains and were co-stained for F4/80 and Osteopontin (*Spp1*). Overview images of the lesioned striatum (str) and the corpus callosum (cc) and corresponding z-stacks are shown. **B** The density of Osteopontin⁺-cells was quantified in eight defined regions 1–6 (covering most of the middle cerebral artery flow area at bregma, see numbered rectangles) generated from brains post 24 h ($n = 6$) and post 72 h ($n = 5$). Data are presented as mean values ± SEM (two-sided *t*-test, *$p = 0.0106$). **C** Heatmaps were generated from brains post 24 h ($n = 6$) and post 72 h ($n = 5$) (two-sided *t*-test, *$p < 0.05$). 3D heatmap of Osteopontin expression was constructed by combining nine consecutive transversal sections of 72 h post-MCAO brains ($n = 3$) using Free-D software. Neuronal-marker MAP2 was used to define lesion borders. **D** Sections of 24 h and 72 h post-MCAO brains as in **A** were co-stained for F4/80 together with LPL, M-CSF, and ADAM8. Representative Z-stacks are shown. The density of positive cells was quantified within the defined sections 1–6 of 24 h ($n = 6$) and 72 h ($n = 4$) post-MCAO brains. Plots in **B** and **D** are representative of $n = 4$ stainings (sham), $n = 6$ stainings (24 h post MCAO) and $n = 4$ stainings (72 h) (two-sided *t*-test, **$p < 0,01$, ***$p < 0.001$; LPL/F4/80: $p = 0.0087$; M-CSF/F4/80: $p = 0.0008$; ADAM8/F4/80: $p = 0.0043$). Data are presented as mean values ± SEM. **E** Mice ($n = 3$) were injected with idoxuridine (IdU) twice 48 h and 24 h prior to induction of MCAO and IdU was co-stained with M-CSF. **F** Sections as in **A** were co-stained with FluoroMylein and SAMC-marker M-CSF, showing potential myelin phagocyting by CD45- and M-CSF-expressing cells. **G** M-CSF-expressing cells potentially phagocyting APC⁺-oligodendroglia. **H** LPL⁺-cell enclosing a NeuN⁺-neuron within the lesioned striatum. **I** Representative immunofluorescence images and corresponding z-stacks of SAMC-marker Osteopontin, LPL, M-CSF, Adam8, and MMP12 co-stained with microglial/monocyte marker IBA1 within the infarcted tissue of stroke patients (representative images of $n = 5$ patients). **J** Human stroke sections stained for SAMC-marker M-CSF and neuronal-marker MAP2 showing potential neuronal phagocyting by M-CSF⁺ cells. Source data for **B** and **D** are provided as a Source Data file.

labels peripheral macrophages as tdTomato⁺, but not microglia in homeostasis and stroke[32]. With this considerably more stringent approach, we found that only a small proportion of M-CSF⁺, MMP12⁺, and Osteopontin⁺ cells in the infarction core were tdTomato⁺Iba1⁺ (Supplementary Fig. 8D–F). This

again suggested mixed resident microglia and peripheral macrophage origin, albeit with predominance of the resident origin. This indicates that the myeloid 'stroke response' is specific to stroke, but independent from the cells' developmental origin.

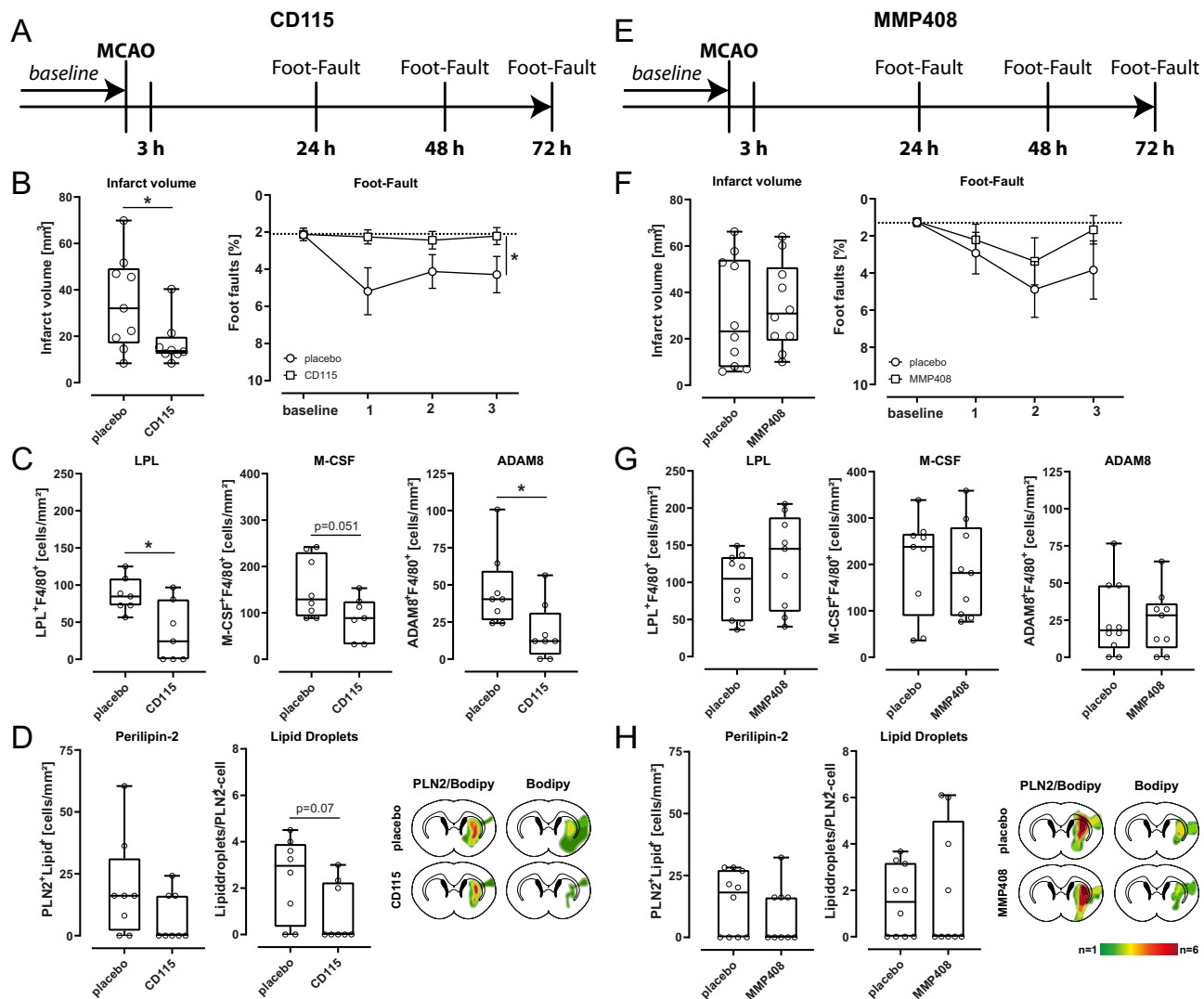

**Fig. 5 Specific SAMC blocking treatment influences stroke outcome and immune cell response. A, E** Schematic illustration of **A** M-CSF antibody (CD115) and **E** MMP-12 inhibitor (MMP408) application over time. Middle cerebral artery occlusion (MCAO) (or sham-operated, as controls) was induced in mice for 45 min. Mice were either injected with anti-CSF-1R mAb (CD115, M-CSF antibody) intraperitoneally at doses of 400 µg/mouse in 100 µl PBS or perorally administered with MMP408 (MMP-12 inhibitor) at doses of 750 µg/mouse in 100 µl PBS 3, 24, and 48 h after induction of MCAO. Functional outcome was assessed with foot fault performances 24, 48, and 72 h after MCAO. Seventy-two hours post-ischemia mice were intracardially perfused and tissue sections were analyzed by immunohistochemistry. **B, F** Mean infarct volumes were calculated from coronal cryosections (15–20) of mice treated with M-CSF antibody or MMP-12 inhibitor in comparison to vehicle-treated animals collected at 300 µm intervals and stained with toluidine blue using ImageJ software at 72 h after MCAO (two-sided $t$-test, *$p < 0.05$, $n = 8$ per group; M-CSF antibody: $p = 0.0360$; MMP-12 inhibitor: $p = 0.3131$). Median infarct volumes were presented as box plots. The lines inside the boxes denote the medians. The whiskers of box plots: 10–90%. The behavioral performances of the mice in each group were measured with the foot fault test at 24, 48, and 72 h post ischemia ($p < 0.05$, two-sided $t$-test, $n = 9$ per group). Data are presented as mean values ± SEM. **C, G** Quantification of LPL+/F4/80+·M-CSF+/F4/80+ and ADAM8+/F4/80+ cells at 72 h post stroke induction after blocking therapy with SAMC-specific markers M-CSF and MMP12 in comparison to vehicle-treated animals (*$p < 0.05$, **$p < 0.01$, two-sided $t$-test, $n = 9$ per group). M-CSF antibody: LPL+/F4/80+: $p = 0.0134$; M-CSF+/F4/80+: $p = 0.051$;ADAM8+/F4/80+ $p = 0.0253$. Median cell counts were presented as box plots. The lines inside the boxes denote the medians. The whiskers of box plots: 10–90%. **D, H** The density of Perilipin-2 and BODIPY positive cells was quantified on cryosections at bregma 0–1 mm 72 h post-MCAO (*$p < 0.05$, two-sided $t$-test, $n = 9$ per group). Cells were quantified within the region of interest (ROI: striatum of the ischemic hemisphere, measuring 0.256 mm²) on coronal sections at 20x magnification, and heatmaps were generated from brains post 72 h ($n = 8$) (two-sided $t$-test, *$p < 0.05$). Median cell counts were presented as box plots. The lines inside the boxes denote the medians. The whiskers of box plots: 10–90%. Source data for **B**, **C**, **D**, **F**, **G**, and **H** are provided as a Source Data file.

**Blocking markers of stroke-associated myeloid cells modulates neuronal injury and functional outcome after rodent stroke.** We next sought to investigate the functional relevance of SAMC in stroke. We, therefore, selected SAMC-specific markers (M-CSF and MMP12) secreted or located on the cell surface and blocked them in mice using monoclonal antibody (anti-M-CSF/CD115) or inhibiting peptide (MMP12/MMP408) (Fig. 5A, E). We then

studied how this influenced infarct volume and functional outcome after 30 min MCAO experimental ischemia. Foot fault performances were significantly improved following treatment with an M-CSF specific antibody compared to vehicle-treated animals (Fig. 5B). Accordingly, infarct volumes were significantly decreased in mice treated with M-CSF specific antibodies compared to vehicle-treated animals (Fig. 5B). The amount of LPL

and ADAM8 proteins within the infarcted area was significantly reduced after the blockade of M-CSF (Fig. 5C). Functional and structural recovery was not significantly influenced after treatment with an MMP12-specific inhibitor (Fig. 5F). However, we could detect a trend towards a better functional outcome after treatment with MMP12-specific inhibitor 72 h after induction of MCAO (Fig. 5F). Furthermore, after treatment with MMP12-specific inhibitor, the SAMC-marker LPL, M-CSF, and Adam8 were not significantly altered in comparison to vehicle-treated animals (Fig. 5G). To further validate the phagocytic function of SAMC, we performed immunohistochemical staining for Perilipin-2 and BODIPY in mice after blockade of SAMC-specific markers (Fig. 5D, H). In all mice treated with SAMC-marker specific antibody/inhibitor, we found a trend towards a decreased amount of Perilipin-2[+] cells (Fig. 5D, H) and part of lipid droplets (Fig. 5D) indicating a reduced phagocytic activity. In summary, we found that antibody-mediated blocking of SAMC-specific markers ameliorated stroke outcome potentially through reduced phagocytosis of cell debris, thus supporting the functional relevance of SAMC in stroke.

## Discussion

We here identify tissue-specific alterations of leukocytes in every CNS-associated border compartment. Compared to the meninges, immune cells in the brain parenchyma feature a delayed and reduced influx of DCs and T cells; likely mirroring the myeloid to lymphocyte influx gradient from the meninges towards the parenchyma described previously[4,6]. Second, using single-cell transcriptional profiling, we identify and subsequently confirm a phenotype which we name stroke-associated myeloid cells (SAMC) instructed onto the myeloid lineage by the ischemic CNS tissue. We find that these cells are not likely an ontogenetically homogeneous population, but are derived from both resident microglia and peripheral monocytes/macrophages with microglial predominance. The core gene signature comprising *Spp1*, *Fabp5*, *Gpnmb*, *Ctsb*, *Ctsl*, *Lgals3*, *Lpl*, *Fth1*, *Cd63*, and *Ctsd* distinctly characterized SAMC. We speculate that the ischemic tissue instructs this phenotype onto myeloid cells of different ontogenetic origin.

Third, a detailed examination of SAMC, showed signs of enhanced lipid metabolism and phagocytosis, suggesting a need for clearance of lipid-rich tissue debris after stroke. It is known that activated microglial cells phagocytose neuronal material after stroke[33,34]. Recent studies also suggest that microglia increase phagocytic activity following stroke, albeit at significantly lower levels than recruited monocytes[35]. In fact, immunostaining of SAMC-specific genes followed by lipid phagocytosis assay identified lipid-phagocytosing activity of SAMC cells. Recently, triggering receptor expressed on myeloid cells 2 (TREM2) was shown to attenuate phagocytic activity of microglia after stroke[34,35]. In detail, TREM2 deficiency induced a reduced tissue resorption with increased infarct size and a worsened neurological outcome. Interestingly, microglial TREM2 expression, and not TREM2 expression on circulating macrophages, was fundamental in stroke outcome[34,35]. We integrate these previous and our findings into a model where SAMC may modulate stroke outcome through phagocytosis of lipid-rich tissue debris.

Remarkably, phagocytosis assays demonstrated that SAMC cells characterized as CD45[high]Lilrb4[+] cells, were much bigger and showed a more intense signal uptake of a lipid dye compared to non-SAMC (CD45[+]Lilrb4[−]) and also SAMC cells expressing a microglia phenotype (CD45[low]Lilrb4[+]). However, these cells did not form as many lipid droplets as expected in highly phagocytosing cells indicating fast processing of lipids as being recently described in receptor-mediated (such as CD36, SR-A1, or Lox1)

phagocytosis of oxidized low-density lipoprotein (Ox-LDL)[36]. In fact, at the sites of atherosclerotic lesions, lipid-droplet-rich foamy macrophages provide decreased phagocytosis activity compared with macrophages without lipid droplets[37]. Interestingly, CD36, a phagocytic receptor that mediates the uptake of fatty acid-containing ligands, is highly expressed in the SAMC population. In stroke, CD36 was found to provide a context-dependent function involving inflammation and resolution with either beneficial and detrimental outcome effects[38]. In conclusion, our findings thus lend support to a phagocytic function of SAMC in stroke in rodents and human affecting both functional and structural outcomes.

Moreover, SAMC were identified to share some gene expression pattern with neurodegenerative disease-associated microglia[13,14] but surprisingly even more with early proliferative-region-associated microglia[39] and axon tract-associated microglia found in embryonal brain[30]. As we observed a partial developmental-like phenotype of SAMC, this indicates that myeloid phenotypes expressed during developmental stages might be reactivated not only in neurodegeneration[39], but also in ischemic stroke. The overlap in gene expression profile might be due to similarities in functional properties such as phagocytosis[14,39].

Rodent stroke models are often limited in understanding human stroke pathology. We aimed to address this issue by performing a power calculation and blinded assessments as recommended in the Stroke Academic Industry Roundtable (STAIR) criteria[40,41]. A potential weakness of our study is that we chose young male mice in order to reduce variables with sex differences due to estrous cycle, and hence to reduce the number of animals used for ethical issues. Furthermore, the use of healthy animals without comorbidities may influence the structural and functional outcome after stroke. We aimed to mitigate this shortcoming by confirming the presence of SAMC phenotype cells in several rodent models of stroke, strains, and across species.

In summary, we have learned that the transcriptional response of myeloid cells in the brain is partially conserved across different types of injury, but still specific enough to define a stroke myeloid gene set. Moreover, the signals which induce the SAMC phenotype remain to be determined. One speculates if secreted stroke-specific markers could be used as stroke biomarkers in human patients. Single-cell transcriptomic studies of patients who suffered from stroke will be needed to further elucidate the function of SAMC in ischemic stroke. The identification and understanding of the utility of CNS and border compartment-associated leukocytes during stroke is essential to developing new therapeutic approaches, specific to previously unidentified subsets.

## Methods

**Ethical regulations.** All animal procedures were performed in accordance with local animal welfare regulations and experimental protocols were approved by the local governmental authorities (Landesamt für Natur, Umwelt und Verbraucherschutz, NRW, Germany, Istanbul Medipol University Animal Research Ethical Committee, and Landesamt für Verbraucherschutz, Thüringen, Germany) under the approval reference number 81-02.04.2018.A316, E-38828770-772.02-3878; 19.08.2021/57 and 02-075/16. Research of human brain autopsy samples was conducted in accordance with the declaration of Helsinki and was approved by the local ethics committees of the Ärztekammer Westfalen-Lippe and Westfälische Wilhelms-University, under reference number 2017-210-f-S. Written informed consent was obtained from all patients or legal representatives. Participant compensation was not offered.

**Animals.** Adult (10–16 weeks of age, $n = 175$) male C57BL/6 mice were used in all experiments, adult Balb/c male mice (10 weeks of age, $n = 15$) and adult male Wistar rats (10–16 weeks of age, $n = 7$) were used for immunohistochemistry, and adult (10–16 weeks of age, $n = 14$) male B6.SJL-Ptprc[a]Pepc[b]/BoyJ (i.e. CD45.1

congenic C57BL/6) mice were used in bone marrow chimera experiments. Cxcr4CreER/WtR26CAG-LSL-tdT mice were generated as follows: A Cxcr4CreER(T2)-IRES-eGFP knockin allele was generated by inserting CreER(T2)-IRES-eGFP into the exon 1 of the Cxcr4 locus. Following this, Cxcr4CreER does not produce functional Cxcr4. Cxcr4CreER males were bred to females of the R26CAG-LSL-tdT Cre-reporter strain Ai1446 and to Cxcr4LoxP; R26CAG-LSL-tdT females, thus generating Cxcr4CreER/wt; R26CAG-LSL-tdTcontrol and Cxcr4CreER/LoxP; R26CAG-LSL-tdT inducible Cxcr4cKO mice[32]. Mice were maintained under specific pathogen free (SPF) conditions on a 12:12 h light-dark cycle period and had access to pelleted food and water ad libitum. Mice were kept in standard housing conditions with ambient temperature between 20–24 °C and humidity between 45–65%. In histological and flow cytometric analysis sham-operated wild-type mice served as controls, in scRNA-seq analyses naive wild-type mice were used as controls.

**Sample size calculation.** Using the sample size calculator available at http://www.stat.ubc.ca, we performed a priori sample size calculations to achieve 80% power to detect a relevant treatment effect of 25% with an alpha level of 0.05.

**Administration of antibody specific for M-CSF (CSF1-R, CD115) and inhibitor specific for MMP-12 (MMP408).** Recipient animals were injected with anti-CSF-1R mAb (αCSF-1R, clone AFS98) intraperitoneally at doses of 400 μg/mouse in 100 μl PBS 3, 24, and 48 h after induction of MCAO (Fig. 5A). The selective inhibitor MMP-12 was administered perorally at doses of 750 μg/mouse in 100 μl PBS 3, 24, and 48 h after induction of MCAO (Fig. 5B). Mice in the control group were treated with the vehicle of peptide (100 μl PBS), using the same protocol.

**Proliferation.** In order to label proliferating cells, C57BL/6 male mice ($n = 3$) received two injections of thymidine analog iododeoxyuridine (IdU, 50 mg/kg/d, i.p.) on day 2 and day 1 prior to induction of 45 min MCAO.

**Bone marrow chimeras.** Bone marrow chimeras were generated via transplantation of $0.5 \times 10^7$ congenic bone marrow cells into the tail vein of sublethally irradiated (7.5 Gy) wildtype recipient mice (C57Bl/6 J CD45.2 in C57Bl/6 J CD45.1 or vice versa). Chimerism was controlled after 8 weeks of reconstitution by staining for allelic CD45 variants within blood samples collected from the tail vein using flow cytometry (CD45.1 vs. CD45.2). Animals with >90% CD45.1$^+$ and <10% CD45.2$^+$ leukocytes or >90% CD45.2$^+$ and <10% CD45.1$^+$ leukocytes were used for subsequent experiments.

**Human samples.** Brain autopsy material of five patients (three women, two men; mean 68.4 years, range: 50–86 years) suffering from acute ischemic stroke who died between 1 and 7 days (median 3 days) after stroke onset at the University Hospital of Münster was analyzed in this study. Samples were derived from brain regions mainly comprising the frontal cortex, striatum, and internal capsule, therefore contained in the vascular territory of the middle cerebral artery. Histopathologically, all analyzed tissue samples were classified as stage I lesions (phase of acute neuronal injury) on adjacent hematoxylin and eosin-stained sections (5 μm distance to analyzed sections) as previously described[42].

**Middle cerebral artery occlusion.** Induction of transient focal experimental stroke was performed by temporary occlusion of the middle cerebral artery (MCAO). Therefore, mice and rats were anesthetized and maintained, under constant body temperature of 37 °C ± 0.5 °C with 1.5% isoflurane in 30% O₂/70% N₂O throughout the procedure. Following the midline neck incision, the left common carotid artery and carotid bifurcation were exposed, and the proximal left common and external carotid arteries were ligated. To transiently interrupt retrograde perfusion of the left common carotid artery a microvascular clip (FE691; Aesculap) was used. MCA-occlusion was performed by insertion of a silicon-coated 8–0 nylon monofilament (701956PK5Re, Doccol Corporation, Sharon, MA) through a small incision of the common carotid artery. After 30 or 45 min of MCAO, monitored by laser Doppler (Periflux 5001; Perimed), the monofilament was withdrawn for reperfusion of the middle cerebral artery and the wound was closed. Sham animals underwent the same procedure, except that the nylon filament was retracted immediately after insertion.

**Induction of focal brainstem ischemia.** Induction of focal brainstem ischemia was performed under general anesthesia achieved by intraperitoneal injections of ketamine hydrochloride and xylazine hydrochloride. During surgery, rectal temperature was monitored and maintained at 37 °C. Following the ventral midline incision, the base of the cranium was approached by blunt separation laterally from the trachea and the esophagus. The bulla tympanica was used as a lateral anatomical landmark. A point exactly between the midline and where the internal carotid artery enters the cranium (ie, the carotid canal) was selected in order to reliably illuminate a defined part of the brainstem through the skull. We used a speculum (diameter, 1.7 mm) attached to a stereotactic frame to prevent occultation by soft tissue. A skull window or thinning of the skull bone has not been necessary since laser penetration depth was adequate to assure illumination of

vessels supplying the brainstem. 1 ml of Rose Bengal at a concentration of 0.133 ml/kg or 1 ml of saline (sham) was injected intravenously over 1 min and laser illumination (wavelength, 560 nm) was applied for 3 min.

**Photothrombotic stroke.** For induction of photothrombotic cortical stroke, rats were anesthetized with 1.5% isoflurane in 30% O₂/70% N₂O. During surgery, a thermostat-controlled heating pad served to maintain a constant body temperature at 37 °C ± 0.5 °C. After a dorsal midline incision, a cold light source (KL1500, Zeiss, Jena, Germany) with a diameter of 4 mm was positioned 3 mm posterior to the bregma and 3 mm right from the midline. Cortical microvessel occlusion was induced by intraperitoneal injection of 0.15 ml Bengal rose (10 mg/ml) followed by skull illumination for 20 min. Sham animals underwent the same procedure, including Bengal rose injection, except illumination of the skull. After the procedure, the skin was sutured, and rats were allowed to recover from anesthesia.

**Functional testing.** The foot fault test was performed to compare sensorimotor deficits. In the foot fault test, the animals were placed individually on an elevated 10-mm square wire mesh with a total grid area of 40 cm × 40 cm. The animals were videotaped while walking freely for 2 min and the number of foot faults and the total number of steps were counted. The percentage of foot faults was calculated as follows: number of foot faults/number of total steps * 100.

**Tissue collection and processing for histology.** Twenty-four and seventy-two hours after MCAO, mice, and rats were perfused through the left ventricle with phosphate-buffered saline (PBS) for 5 min followed by 4% paraformaldehyde solution for 10 min under deep xylazine/ketamine anesthesia. Brains were removed, fixed in 4% paraformaldehyde overnight, immersed in 20% sucrose for 3 days, frozen, and stored at −80 °C.

**Infarct volume assessment.** Calculation of infarct volume was done by collecting coronal cryosections (10 μm) every 300 μm starting at the rostral border of the infarct. Slices were stained with 0.5% toluidine blue (Sigma, St Louis, MO) followed by drying in graded ethanol for 1 min each (50, 80, 96, and 100%). Digitized images of infarct area and the areas of ipsilateral hemisphere and contralateral hemisphere were measured by a blinded investigator on each section using ImageJ software 1.48 v. The final infarct volume was calculated by multiplying infarct area size by the distance to the next section. To compensate for edema formation the formula (area of contralateral hemisphere/area of ipsilateral hemisphere) * infarct area was applied.

**Immunohistochemistry.** Mounted coronal mouse cryosections were rinsed in 3% H₂O₂/Methanol for 10 min to block endogenous peroxidases and thereafter incubated in Blocking Reagent (Roche Diagnostics) for 15 min to prevent nonspecific protein binding. We used the following primary antibodies for murine sections: Rabbit-anti-NeuN (1:150, clone 27-4, Millipore, MABN140), Mouse-anti-GFAP (1:500, clone G-A-5, Millipore, G3893), Rat-anti-Ly-6B.2 (1:100, clone 7/4, BioRad, MCA771G), Rat-anti-F4/80 (1:500, clone CI:A3-1, Serotec, MCA497G), Hamster-anti-CD3 (1:50, BDBioscience, 550277), Goat-anti-Iba1 (1:50, Abcam, ab5076), Rat-anti-B220/CD45R (1:100, clone RA3-6B2, ThermoFisher, 14-0452-82), Rabbit-anti-Laminin (1:100, Abcam, ab11575), Rabbit-anti-Osteopontin (1:100, Abcam, ab63856), Mouse-anti-LPL (1:100, clone LPLA4, Abcam, ab21356), Rabbit-anti-Tspan8 (1:100, LSBio, LS-B11508), Mouse-anti-APC (1:100, Merck, OP80), Rabbit-anti-CSF1/M-CSF (1:100, clone ERP20948, Abcam, ab233387), Rabbit-anti-MMP12 (1:100, Abcam, ab128030), Rabbit-anti-MS2/Adam8 (1:100, clone EPR22688-44, Abcam, ab255608), Goat-anti-FABP5 (1:100, R&D Systems, AF1476), Chicken-anti-MAP2 (1:500, Abcam, ab5392), BODIPY 493/503 (1:1000, Molecular Probes, D3922) and Rabbit-anti-Perilipin-2 (1:100, Novusbio, NBP2-48532). Subsequently, brain slices were incubated with the appropriate Alexa Fluor secondary antibody for antigen visualization (1:100, 45 min, room temperature). Cellular nuclei were counterstained using a fluorescent preserving mounting medium containing 4',6-diamidino-2-phenylindole (DAPI, Life, 00-4952-52). To amplify the signal of F4/80 we applied HRP-conjugated streptavidin (DAKO, Denmark, 1:100, 45 min) and biotinyl tyramide (1:100, 15 min), after incubation with respective biotinylated secondary antibodies (biotinylated anti-rat antibody, 1:100). Afterward, amplified antigens were visualized with streptavidin-conjugated dye (Alexa Fluor594, Molecular Probes, 1:100, 45 min). Apoptotic cells were stained with terminal deoxynucleotidyl transferase dUTP nick-end labeling (Click-iT Plus TUNEL Assay, Thermofisher Scientific, C10617). For labeling of proliferation marker IdU, brain slices were pre-treated in 50% saline sodium citrate buffer (SSC)/50% formamide (2 h, 60 °C). After washing in SSC, DNA was made accessible by incubation in 0.65 N HCl (30 min, 60 °C) followed by thorough washing in 0.1 M borate buffer (pH 8.5, 10 min, room temperature). After Fc-blocking with Blocking Reagent (Roche, 10% goat serum added), IdU$^+$-cells were immunostained with Mouse-anti-IdU antibody (1:200, clone B44, BD Pharmingen, BD347580). Myelin was stained using FluoroMyelin Green (1:100, 20 min, Thermofisher, F34651).

Human brain sections were rehydrated and antigens unmasked (EnVisionTM FLEX Target Retrieval Solution, pH 9.0) under heat. After washing (PBS, 0.02% Tween 20), sections were incubated in PBS (10% FBS, 20 min) and stained using

the following primary antibodies: Rabbit-anti-Osteopontin (1:100, Abcam, ab63856), Mouse-anti-LPL (1:100, clone LPLA4, Abcam, ab21356), Rabbit-anti-CSF1/M-CSF (1:100, clone ERP20948, Abcam, ab233387), anti-MS2/Adam8 (1:100, clone EPR22688-44, Abcam, ab255608), Rabbit-anti-MMP12 (1:100, Abcam, ab128030), Chicken-anti-MAP2 (1:500, Abcam, ab5392), Goat-anti-FABP5 (1:25, R&D Systems, AF1476) and Goat-anti-IBA1 (1:50, Abcam, ab5076). All stainings were mounted with Vectashield Mounting Medium with DAPI (Life, 00-4952-52). Images were taken with a Nikon Eclipse 80i fluorescence microscope (Nikon) and a Zeiss AxioVision Apotome (Carl Zeiss). ImageJ software 1.48 v was used for manual cell counting. HeatMaps of immune cell infiltration were drawn using Adobe Illustrator (Adobe Illustrator CS5).

**Blinded assessment**. Analysis of functional testing, infarct volume assessment, and quantification of histological findings was performed in a blinded manner by a single experienced medical technical assistant and reviewed by JKS.

**3D-visualization**. Three-dimensional visualization of Osteopontin expression was done with Free-D software[43]. Reconstruction was performed using 9 consecutive transversal brain slices with a spacing of 200 μm stained for Osteopontin and MAP2. Neuronal MAP2-staining was used as a marker for damaged CNS tissue.

**Isolation of CNS-resident leukocytes**. Mice were injected (intravenously; iv) with fluorochrome coupled anti-mouse CD45 antibody (clone 30-11 F, Biolegend, 3 μg/mouse). Five minutes later, animals were intracardially perfused with cold PBS with 10 U/ml heparin under deep ketamine/xylazine anesthesia. During perfusion, blood was collected and transferred into a tube. To separate different CNS compartments, the forebrain and cerebellum were dissected from mice. Under binoculars, the dura was dissected from the calvaria and the pia was dissected from the brain parenchyma. All choroid plexuses were removed from the ventricles of the brain. Brain tissue was cut into pieces and digested with collagenase D (2.5 mg/ml, Roche Diagnostics) and DNase I (0.05 mg/ml, Sigma) at 37 °C for 20 min. Pia, dura, and CP were digested in medium containing 1 mg/ml Collagenase D (Roche) for 45 min at 37 °C. Afterward, digested tissue was passed through a 70 μm cell strainer. The parenchyma was centrifuged on a 70%/37% Percoll gradient. The interphase was removed, washed, and resuspended in PBS with 2% FCS. All samples were processed by staining and flow cytometry or flow sorting.

**Flow cytometry and sorting**. The following fluorochrome-labeled antibodies were used: CD45 (30-11 F) BV510 or FITC 1:100, CD45R/B220 (RA3-6B2) PerCP-Cy5.5 1:100, CD3 (17A2) PE-Cy7 1:200, F4/80 (BM8) APC 1:200, Ly-6G/Ly-6C (RB6-8C5) BV421 1:200, CD11c (N418) AF700 1:150, CD11b (M1/70) PE 1:800 all obtained from Biolegend and NK-1.1 (PK136) APC-Vio770 1:200 obtained from Miltenyi Biotec. After staining, cells were washed twice and resuspended in PBS with 2% FCS. Cells were acquired on a Gallios flow cytometer (Beckman Coulter) or sorted on a FACS Aria III (BD). Sorting was performed using an 85 μm nozzle and 4-way purity sort precision mode. Data were analyzed using FlowJo software v10.6.1 (BD). Cells extracted from the CNS, pia, and dura were sorted for CD45$^+$CD45iv$^-$ cells (Supplementary Fig. 2). Cell concentrations from all tissues were manually counted in a Fuchs-Rosenthal counting chamber. Sorted single-cell suspensions were used for subsequent single-cell RNA-sequencing.

**Generation of single-cell libraries and sequencing**. Single-cell suspensions were loaded onto the Chromium Single Cell Controller using the Chromium Single Cell 3′ Library & Gel Bead Kit with v2 and 3 chemistry (both from 10X Genomics, #1000128, and #1000157). Sample processing and library preparation was performed according to manufacturer's instructions using AMPure beads (Beckman Coulter, A63881). Sequencing was carried out on an Illumina Novaseq using NovaSeq 6000 S4 Reagent Kit (300 Cycles, #20028312) with a 150-8-8-150 read setup. Details regarding sequencing depth and cell recovery are provided in Supplementary Data 1.

Processing of sequencing data was performed with the cell range pipeline v6.0 (10X Genomics) according to the manufacturer's instructions. Briefly, raw bcl files were de-multiplexed using the cellranger mkfastq pipeline. Subsequent read alignments and transcript counting was done individually for each sample using the cellranger count pipeline with standard parameters.

**Single-cell RNA-sequencing analysis**. Downstream analysis was performed with the R-package Seurat v4.0.4[44] using R v4.1.1. as previously described[45]. Low-quality cells and cell doublets were removed by filtering cells with few genes (<200), high number of genes (>1500–3000) or high mitochondrial percentages (>4–15%) for each sample separately. Doublets were removed using scDblFinder v1.6 with default parameters. The total remaining cell number used for further analysis was 33,559 (Supplementary Data 1). Data were then merged again and normalized using SCTransform[46].

Principal component analysis (PCA) was applied for primary dimensionality reduction and an elbow plot was used to choose the appropriate number of PCs for further analysis (40 PCs in our dataset). To correct for batch effects, cells were aligned between samples with Harmony[47]. Clusters were identified by the

"FindNeighbors" and "FindClusters" (Louvain method, resolution 0.4) functions in Seurat. To visualize the results, Uniform Manifold Approximation and Projection (UMAP) was performed with Harmony embeddings. Differentially expressed genes were calculated with the "FindMarker" function in Seurat (Wilcoxon rank-sum test). We queried the top differentially expressed genes of each cluster in a literature search to annotate the clusters. Differentially expressed genes were visualized in a volcano plot using the EnhancedVolcano package. We compared cells from SAMC cluster with the mouse cell atlas (MCA) dataset[29] using the scMCA tool[48]. Gene score plots were created with the "AddModuleScore" function in Seurat.

To perform gene set enrichment analysis, we used Enrichr[49,50]. We selected the following reference datasets of Enrichr: TF (transcription factor) Perturbations Followed by Expression, Transcription Factor PPI (protein-protein interactions), Enrichr Submission TF Gene Cooccurrence Enrichment Analysis, WikiPathways 2019 Mouse, KEGG 2019 Mouse, Reactome 2016, and Panther 2016. The enrichment analysis was performed as previously described[49].

**Cxcr4$^{CreER}$-mediated identification of hematopoietic stem cell (HSC)-derived cells in the brain**. The Cxcr4$^{CreER/Wt}$R26$^{CAG-LSL-tdT}$ model was used to discriminate HSC-derived macrophages from microglia[32]. Adult Cxcr4$^{CreER/Wt}$R26$^{CAG-LSL-tdT}$ mice received tamoxifen (5 × 1 mg intraperitoneally) to label HSCs. Mice underwent 28 days tamoxifen washout before subsequent experiments. Recombination efficiency was assessed for circulating CD11b$^+$CD115$^+$ Ly6C$^+$ monocytes by flow cytometry and exceeded 86% in all mice. Mice underwent 45 min MCAO and were processed for immunohistochemistry at days 1, 2, and 3 after MCAO induction (n = 3 each). HSC-derived macrophages were defined as tdTomato$^+$Iba1$^+$ and microglia as tdTomato$^-$Iba1$^+$-cells. Antibodies for co-staining in this model were: rat anti-RFP (Chromotek 5F8, 1: 500), goat anti-Iba1 (Abcam ab5076, 1: 500), rabbit anti-Osteopontin (Abcam ab63856, 1:2000), rabit anti-MCSF (Abcam ab233387, 1:2000), and rabbit anti-MMP12 (Abcam ab128030, 1:2000). Rabbit antibodies were detected using the biotin/tyramine amplification procedure and Streptavidin-AF488. Percent overlap of microglia and HSC-derived macrophages with M-CSF, Osteopontin, and MMP12 was examined in the striatal infarct in three 40x confocal view-fields per animal captured with an LSM900 microscope (Zeiss).

**Lipid phagocytosis assay**. Sorted cells (around 8 × 10³ cells for CD45$^{low}$ Lilrb4$^+$ and CD45$^{high}$ Lilrb4$^+$ subsets and 2 × 10⁴ for CD45$^+$ Lilrb4$^-$ subset) were collected after centrifugation and distributed in two wells, for control cells and cells fed with Ox-LDL, of a chambered coverslip with 8 wells (Ibidi Cat No. 80826) for cell culture and immunofluorescence, in 250 ml per well of RPMI 1640 supplemented with 10% FCS 1% L-Glutamine and 1% Penicillin/Streptomycin. For lipid phagocytosis assay, cells were incubated for 18 h in the presence of 25 mg/ml oxidized-LDL (Invitrogen Cat No. L34357) and then washed once with PBS by centrifugation, in order not to lose the non-adherent cells. For lipid droplets visualization due to ox-LDL ingestion, cells were incubated with 1 mg/ml of Bodipy 493/503 solution (Invitrogen Cat No. D3922) for 30 min at 37 °C, to then visualize under the microscope by using mounting media containing DAPI. Images were taken at 40x magnification.

**Statistic and reproducibility**. Statistical analysis was performed using R version 4.1.1 and GraphPad Prism version 8 (GraphPad Software, La Jolla, CA). Data were checked for normal distribution applying the Shapiro-Wilk normality test followed by group comparison using the Student's t-test or Wilcoxon rank-sum test. The statistical tests employed for the single-cell RNA-sequencing analysis are integrated in the R packages and are given above. All of the experiments were reproduced at least in duplicate. Data are presented as mean ± SEM. A p-value of <0.05 was considered significant.

**Reporting summary**. Further information on research design is available in the Nature Research Reporting Summary linked to this article.

## Data availability

The raw scRNA-seq data of this study including cluster and sample annotations are available in the GEO repository (GSE189432). Technical details about the sequencing samples as well as the results of the differential expression data enrichment analysis are given in the method section and as supplementary data. Additional source data underlying Figs. 1,4, and 5 and Supplementary Figs 1,6,7 and 8 are provided in the source data file. Source data are provided with this paper.

## Code availability

The processed and annotated single-cell RNA-seq dataset can be interactively explored at: https://osmzhlab.uni-muenster.de/shiny/cerebro_stroke/. The code was based on the official tutorials of the packages listed, no custom code was generated

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

## Acknowledgements

We thank Anna-Lena Börsch, Birgit Schmeddes, and Maike Hoppen for excellent technical assistance. This work was supported by the Interdisciplinary Center for Clinical Research (IZKF) Münster, grant Min3/003/21, and by the Deutsche Forschungsgemeinschaft (DFG; MI 1547/3-1, MI 1547/4–1 and FOR 2879/1). G.M.z.H. was supported by grants from the Deutsche Forschungsgemeinschaft (DFG, grant number ME4050/4-1) and DFG grant number ME4050/8-1, under the frame of E-Rare-3, the ERA-Net for Research on Rare Diseases, by the Grant for Multiple Sclerosis Innovation (Merck), and from the Ministerium für Innovation, Wissenschaft und Forschung (MIWF) des Landes Nordrhein-Westfalen.

## Author contributions

Conception and design of the study: C.B., D.S., J.-K.S., G.M.z.H., J.M. Acquisition and analysis of data: C.B., D.S., J.K.S., M.H., X.L., J.W., A.S.-P., C.T., T.K., I.A.-P., N.A.-G., P.A.K., Y.W., E.K., D.M.H., H.W., R.S., G.M.z.H., J.M.. Drafting manuscript and figures: C.B., D.S., J.K.S., M.H., G.M.z.H., J.M. Revision and approval of manuscript: C.B., D.S., J.K.S., M.H., X.L., J.W., A.S.-P., C.T., T.K., I.A.-P., N.A.-G., P.A.K., Y.W., E.K., D.M.H., H.W., R.S., G.M.z.H., J.M.

## Funding

## Competing interests

The authors declare no competing interests.
