## [Peer Review File · Nature Communications]

Stroke induces disease-specific myeloid cells in the brain parenchyma and piaREVIEWER COMMENTS

Reviewer #1 (Remarks to the Author):

Title: Stroke induces disease-specific myeloid cells in the brain parenchyma and pia

Beuker et al 2020

Main Statement:

The goal of the study was to establish a census of leukocyte populations in the CNS and its associated border compartment regions in a mouse model of stroke. The authors hypothesized that by using this broad census, they would then be able identify and characterize unique immune cell populations that are potentially specific for stroke pathology, as well as gain insight into the shifting dynamics of immune cell populations following stroke in different compartments of the CNS. According to the conclusions made by the authors, their main finding is that they have identified a subpopulation of myeloid cells following stroke, with unique expressional profile designated "stroke associated myeloid cells" (SAMC). The authors then investigated localization, function, and development of these SAMC, concluding that they may have unique phagocytic properties specialized for clearing damaged neuronal tissue. Combined, the three main contributing authors have previous publications in stroke pathology and single cell analysis of leukocytes in inflammatory disorders, and in addition the affiliated labs are well-known for their contributions to understanding the immune response following stroke and other neuropathological diseases. Together their work offers promising insight into stroke specific responses, in the early stages of ischemic injury.

Overall Significance Statement

While this study is not particularly novel in its underlying experimental design (similar methodology have been applied in other disease states), it does use state-of-the-art techniques in scRNA-seq to reaffirm many widely held conclusions about the underlying dynamics and features of leukocyte populations in the CNS following stroke, and offers additional insight into the compartmentalization of immune cells before and after stroke response. In addition, the study builds upon the emerging research that is attempting to phenotype novel disease associated immune cell populations.

Much of the flow analysis regarding the compartmentalization of immune cell migration in the parenchyma are already largely known, and merely confirms our understanding of immune cell migration following stroke. For example, they reaffirm temporal infiltration gradient of myeloid to lymphocyte lineage following stroke, with lymphocytes infiltrating at a later time period. One interesting insight from this paper is the focus on myeloid lineage cells in meningeal compartments of pia and dura. Here the authors confirmed that in the pia and dura myeloid lineage cells increased 24hr following stroke, followed by a secondary wave of lymphocytes at 72hrs. This response differs from the parenchyma however due to the cell types involved, with the pia having an earlier DC influx when compared to parenchyma, and more T-cell increase in both meningeal compartments at 72hrs. Together this data is seemingly noteworthy as it shows a compartment specific response to stroke at 24 & 72 hrs, while also mirroring the myeloid to lymphocyte influx gradient that is expected in the CNS.

The single cell analysis of these regions before and after stroke is noteworthy, and appears to be the first of its kind in stroke. Figure 2, and its associated databases would provide stroke researchers a wealth of information regarding the early influx and changing dynamics of CNS associated immune cells. Outside of the classified "SAMCs", there appear to be several noteworthy avenues of investigation, particularly in the shifting abundance and expressional patterns of mDC1, mDC2, CD8Tc, and Micro1 in the varying regions of compartmentalization. However, since tissue collection occurred 24hrs after stroke, the investigators focused on the early phenotypic shifts that occur in CNS myeloid cells.

These expressional characteristics of these cells fit into an already established framework of CNS-

resident macrophages/microglia that change depending on disease state. This strengthens the investigators argument, as this cell population can then be compared to future studies as researchers begin to identify more important cell populations in a variety of conditions/diseases. SAMCs appear to have important expressional patterns/markers that have been shown previously to be relevant for stroke pathology, but overall the main conclusions of the paper do not necessarily challenge our understanding of myeloid cell dynamics and act to apply an existing framework to stroke.

Summary:

Main Strengths:

- The succession of experiments and analyses following the scRNA-seq experiment are logical and novel in identifying and characterizing the SAMC. Follows a clear scientific approach.
- Complete single cell analysis and dataset would provide scientific community new avenues of investigation regarding immune cell dynamics in early post-stroke period. For example, future studies can compare expressional patterns of SAMC with their own classified cell populations
- Presence of important stroke relevant expressional changes in osteopontin and Lpl in SAMCs is promising for pathological relevance.
- Presence of SAMC markers in both human and mice ischemic tissue is promising for clinical relevance of these cell types, and provides future researchers with targets of investigation
- Close association with degeneration associated microglia (DAM), and differences with IAM and BAM, indicate that these cells likely lie on a spectrum of disease associated activation, and their further characterization could prove useful in establishing a clearer picture of how myeloid cells change function depending on conditions.

Main Weaknesses:

- In some of the compartmentalization results there appears to a large deviation in immune cell migration/response following stroke particularly in monocyte/macrophage, microglia, DC, and granulocyte cell populations. While the results may be sound, interpretation into significance is difficult when there appears a lot of error. Take for example microglia populations in the ischemic CNS parenchyma following stroke. In addition, in figure 1C, there is no error bars displayed on the cell count line graphs, so true interpretation is difficult.
- Cell numbers for input into scRNAseq are low – needs further confirmation of significance
- It is already known that myeloid/microglia populations shift expressional patterns after tissue injury, and scRNAseq analysis would likely cluster these new populations as new cell type. So, the finding is expected, and would be more noteworthy if this was not the case. With that said, the data fits in nicely with current understanding of shifting myeloid cell phenotypes in inflammatory states.
- Significance and necessity of the CNS compartmentalization flow cytometry experiment and relevance to the SAMC phenotyping experiments is unclear and less accessible. Results appear slightly out of place within SAMC narrative. Although characterization of early infiltrating myeloid populations in dura is interesting, the majority of SAMCs appear to be of resident origin, likely forming from a pre-existing cell population. For example, osteopontin-positive SAMCs (one of the main functional markers identified) appear to all be nearly all resident derived.
- No true functional assay of the SAMC, only characterization of their functionality based on IHC markers and expressional changes. Their importance is only hinted at through references and association, and never truly demonstrated to have direct impact on stroke pathology/outcome/etc.

- The term "Stroke associated myeloid cells" will prove to be controversial, as this cellular phenotype may only apply to this specific stroke model (animal, timeframe, methodology, etc). It remains to be seen if these SAMC signatures are truly held constant across all ischemic stroke models, and are relevant to human pathology. While the author's showed that the presence of these SAMC markers exist in human brain stroke samples, without further studies and perhaps a true single-cell characterization of leukocytes across human stroke patients, we should be hesitant to broadly classify these as purely "stroke associated" as there is significant overlap with other degeneration associated phenotypes. This naming of phenotypical shifts in immune cell populations however is not uncommon in the field, but the name SAMC may be prove to be too broad of a classification.
- 24hr timepoint for the scRNAseq tissue collection may be too early and narrow of timepoint to gain a complete accurate picture of SAMC phenotype shift. For example, the presence of SAMC marker positive cells in the parenchyma is much higher at 72hr across all markers investigated (Figure 4), indicating that later timepoints may be more important for the purposes of SAMC classification, not 24hr.

Minor Adjustments:

- Figure 1B/C – Significance markers would be useful
- Figure 1C – Color pallet makes interpretation difficult, consider different more easily differentiable line configurations.
- Figure 1C – smaller immune cell populations (per cell count) are difficult to differentiate, consider alternative methods to display data. For example, supplemental figure 3B.
- Page 5 (Line 13) – Misspelled stroke as: "Stoke"
- P.2 line 5, p4: "deeply characterized" should be rephrased (performed in-depth characterization?)
- Abstract: SAMC should not be abbreviated here
- p12 should say that this is a transient MCAO model and they should include further details regarding following the STAIR criteria for stroke studies (number of mice seems to be low -4-6-, unbiased analysis, justification for males, etc)
-

Conclusion:

Overall while the execution of the study has some faults, for instance the lack of a true functional assessment of SAMCs, in total the scRNAseq analysis does provide researchers with a novel insight into the shifting immune cell dynamics within the different compartments of the CNS. Additionally, while the importance of SAMCs and the accuracy of their classification remains to be seen, the researchers made the initial first steps in associating and differentiating this cluster defined myeloid cell population within the context of both stroke and other pathological states. In summary, the study does not challenge our understanding of disease specific myeloid cells but instead builds upon our understanding of myeloid cell dynamics that has been amassed from other models of CNS disease.

Reviewer #2 (Remarks to the Author):

What are the noteworthy results?

In this study Beuker and colleagues focus on a highly topical area, namely the immune cell landscape in barrier sites of the brain. This area has drawn increasing interest across multiple disease areas in recent years, aided by major advances in technology, including the ability to much better differentiate cell subtypes through single cell transcriptomics etc.

The most noteworthy finding is the demonstration of a specific myeloid population within the barrier structures (pia) of the brain and in the CNS parenchyma that is largely derived from brain resident microglia. The authors define this population as stroke-associated myeloid cells (SAMC). Through detailed expression analyses it is further demonstrated that this cell population has a phagocytotic phenotype and the authors speculate this may be geared towards the clearing of myelin debris resulting from the death of neurones as a result of the stroke. However, no functional data is provided to substantiate this hypothesis, although immunofluorescence staining supports the concept. SAMC are also shown to express osteopontin, which the authors suggest could indicate a later pro-regenerative role for these cells, though this is very speculative and again not supported by functional data. The inclusion of data from post-mortem stroke brains to support the findings in the mouse model is noteworthy and strengthens the translational relevance of the work.

Will the work be of significance to the field and related fields? How does it compare to the established literature? If the work is not original, please provide relevant references.

Unlike Alzheimer's disease and other chronic neurodegenerative conditions, there is a relative lack of studies in stroke that perform detailed analysis of immune cell subsets. This study therefore provides important new insight, albeit largely descriptive. Also, a portion of the data presented largely confirms previous work describing the temporal and spatial changes in the major immune cell subsets post-stroke. It is the presence of the previously un-described SAMC that is the original finding in this work.

Does the work support the conclusions and claims, or is additional evidence needed?

The conclusions are largely supported by the data, albeit the authors are speculative about potential functions of SAMC, without providing direct mechanistic evidence. Also, young male mice are used throughout. Stroke is largely an age-associated disease with co-morbidities. It is now well described that these co-morbidities can significantly influence the 'baseline' inflammatory status and subsequent response to stroke. This is not mentioned by the authors and should at least be acknowledged in the discussion.

Are there any flaws in the data analysis, interpretation and conclusions? Do these prohibit publication or require revision?

Overall the study is well performed with appropriate analyses and interpretation of data.

Is the methodology sound? Does the work meet the expected standards in your field? Is there enough detail provided in the methods for the work to be reproduced?

In some experiments the sample size is low and the authors should provide details of how the number of animals used was determined.

Reviewer Comments and Response

Reviewer 1:

1. The goal of the study was to establish a census of leukocyte populations in the CNS and its associated border compartment regions in a mouse model of stroke. The authors hypothesized that by using this broad census, they would then be able identify and characterize unique immune cell populations that are potentially specific for stroke pathology, as well as gain insight into the shifting dynamics of immune cell populations following stroke in different compartments of the CNS. According to the conclusions made by the authors, their main finding is that they have identified a subpopulation of myeloid cells following stroke, with unique expressional profile designated “stroke associated myeloid cells” (SAMC). The authors then investigated localization, function, and development of these SAMC, concluding that they may have unique phagocytic properties specialized for clearing damaged neuronal tissue. Combined, the three main contributing authors have previous publications in stroke pathology and single cell analysis of leukocytes in inflammatory disorders, and in addition the affiliated labs are well-known for their contributions to understanding the immune response following stroke and other neuropathological diseases. Together their work offers promising insight into stroke specific responses, in the early stages of ischemic injury.

Reply: We thank the reviewer for the positive evaluation of our study.

2. While this study is not particularly novel in its underlying experimental design (similar methodology have been applied in other disease states), it does use state-of-the-art techniques in scRNA-seq to reaffirm many widely held conclusions about the underlying dynamics and features of leukocyte populations in the CNS following stroke, and offers additional insight into the compartmentalization of immune cells before and after stroke response. In addition, the study builds upon the emerging research that is attempting to phenotype novel disease associated immune cell populations.

Reply: We thank the reviewer for the positive evaluation of our study and for raising this important issue. Indeed, the experimental design of our study is not entirely new. However, by using scRNA-seq we were able to elucidate novel insights in the post-ischemic immune cell response.

3. Much of the flow analysis regarding the compartmentalization of immune cell migration in the parenchyma is already largely known, and merely confirms our understanding of immune cell migration following stroke. For example, they reaffirm temporal infiltration gradient of myeloid to lymphocyte lineage following stroke, with lymphocytes infiltrating at a later time period. One interesting insight from this paper is the focus on myeloid lineage cells in meningeal compartments of pia and dura. Here the authors confirmed that in the pia and dura myeloid lineage cells increased 24hr following stroke, followed by a secondary wave of lymphocytes at 72hrs. This response differs from the parenchyma however due to the cell types involved, with the pia having an earlier DC influx when compared to parenchyma, and more T-cell increase in both meningeal compartments at 72hrs. Together this data is seemingly noteworthy as it shows a compartment specific response to stroke at 24 & 72 hrs, while also mirroring the myeloid to lymphocyte influx gradient that is expected in the CNS.

Reply: We thank the reviewer for this comment. First, we have increased the n numbers in order to enhance the informative value of our findings and changed the way of presentation as it was misleading and exaggerated some of the changes. We have further adapted the

discussion of the manuscript to better reflect the compartment specific response to stroke at 24 and 72 hrs. We now discuss this in more detail on pages 4, 5 and 11 of the manuscript as follows: ‘At 24h and 72h after ischemia, we found a significant increase of DCs, monocytes/macrophages and granulocytes in the brain parenchyma over time while the proportion of microglia of all leukocytes decreased probably due to the influx of leukocytes. T cells only started to increase from >24h on, while NK cells remained unchanged. B cells showed a non-significant tendency towards increasing numbers over time as well (Fig. 1B). We next extended our analysis to border compartments with special focus on the meninges (Fig. 1B). As we and others have previously shown ^{4,6}, tissue-resident leukocytes in the *healthy* meninges and the other border compartments were distinct from the parenchyma and overall the responses to ischemia were tissue-specific. In all tissues, B cells and T cells were essentially unchanged while monocytes/macrophages showed an increase from 24h on in the CP (choroid plexus). NK cells were decreased at 72h in the CP (Fig. 1B) while granulocytes show a temporary increase at 24h in the pia. In the dura, DCs significantly decreased over time post-ischemia, while NK cells showed a decrease at 24h. Granulocytes increased in the pia at 24h (Fig. 1B; Suppl. Fig. 2,3).’ and We here *first* identify tissue-specific alterations of leukocytes in every CNS-associated border compartment. Compared to the meninges, immune cells in the brain parenchyma feature a delayed and reduced influx of DCs and T cells; likely mirroring the myeloid to lymphocyte influx gradient from the meninges towards the parenchyma described previously ^{4,6}.’.

4. In some of the compartmentalization results there appears to be a large deviation in immune cell migration/response following stroke particularly in monocyte/macrophage, microglia, DC, and granulocyte cell populations. While the results may be sound, interpretation into significance is difficult when there appears a lot of error. Take for example microglia populations in the ischemic CNS parenchyma following stroke. In addition, in figure 1C, there is no error bars displayed on the cell count line graphs, so true interpretation is difficult.

Reply: We thank the reviewer for this important comment. Comparable to human stroke, the middle cerebral artery occlusion model which we used in our study features a considerable degree of interindividual variability, low cell numbers per small border tissues, and thus large error bars. To address this, we have performed a power calculation and found that 12 animals would reasonably be required to detect significant differences presuming a two-sided test, an alpha of 0.05. We have therefore considerably extended the animal numbers in all relevant experiments and essentially tripled the number of biological repeats (previously n = 4-5 animals; now n = 12 animals per MCAO timepoint). However, our initial data regarding spatiotemporal immune cell distribution in the post-ischemic brain have not changed significantly but we were able to reduce data variability. We also attempted various other depictions of our data (previously % of each cell type and absolute numbers of each cell type) and now depict data as scatter plots in % of each cell subtype over time in Figure 1 and Supplementary Figure 3. as we think that this presentation is more suited. We hope to thereby facilitate interpretation of our results in the context.

5. Cell numbers for input into scRNAseq are low – needs further confirmation of significance

Reply: We thank the reviewer for this constructive suggestion. We fully agree with the reviewer that our findings needed further verification. We therefore repeated tMCAO in an independent set of mice at 24 and 72 hours and analyzed 32,457 sorted leukocytes by scRNA-seq. This increased the cell number of the CNS parenchyma with the most relevant

observations (9,807 additional cells). Of note, in comparison to other recent papers (Ochocka et al., NatCommun 2021; Geirsdottier et al., Cell 2019; Brioschi et al., Science 2021), the total amount of cells processed for scRNA-seq (32,457 total cells) is not as unusually low but rather above average. By increasing the cell numbers and analyzing a second time point we were able to systematically compare the leukocyte compositional changes as follows: ‘In a 24h vs. 72h post-ischemia comparison, Granulo_1, Macro_2 and CAM_2 clusters increased, while Microglia clusters (Micro_1, Micro_2, Micro_3 and stress_Micro) and CAM_1 decreased (Fig. 2C,D). Interestingly, the CAM_2 cluster increased more strongly from 24 to 72h post-ischemia than from ctrl to 24h (Fig. 2C). These results may reflect second wave immune responses towards more specialized granulocytes and specific Macrophage/CAM populations.’

6. It is already known that myeloid/microglia populations shift expressional patterns after tissue injury, and scRNAseq analysis would likely cluster these new populations as new cell type. So, the finding is expected, and would be more noteworthy if this was not the case. With that said, the data fits in nicely with current understanding of shifting myeloid cell phenotypes in inflammatory states.

Reply: We thank the reviewer for this comment and fully agree that the phenotype we denote as stroke-associated myeloid cells (SAMC) is likely not a distinct population but rather a specific phenotype instructed onto the myeloid lineage by the ischemic CNS tissue. SAMC are not likely an ontogenetically homogeneous population. In fact, we present evidence (Suppl Fig. 8) that these cells are derived from both resident microglia and peripheral monocytes/macrophages. We now emphasize this aspect in the text and discuss it in more detail in the discussion section on page 11 of the manuscript as follows: ‘Second, using single cell transcriptional profiling, we identify and subsequently confirm a novel phenotype which we name stroke-associated myeloid cells (SAMC) instructed onto the myeloid lineage by the ischemic CNS tissue. We find that these cells are not likely an ontogenetically homogeneous population, but might be derived from both resident microglia and peripheral monocytes/macrophages. The core gene signature comprising *Spp1*, *Fabp5*, *Gpnm3*, *Ctsb*, *Ctsl*, *Lgals3*, *Lpl*, *Fth1*, *Cd63* and *Ctsd* distinctly characterized SAMC. We speculate that the ischemic tissue instructs this phenotype onto myeloid cells of different ontogenetic origin.’ Moreover, our data demonstrate that in comparison to other macrophage clusters, SAMC uniquely express genes that are known to be involved in lipid metabolism and phagocytosis of myelin. In detail, SAMC providing a macrophage phenotype (CD45^{high}Lilrb4⁺) show a more intense signal uptake of a lipid dye BODIPY compared to non-SAMC macrophages (CD45⁺Lilrb4⁻) (Suppl. Fig. 6D).

7. Significance and necessity of the CNS compartmentalization flow cytometry experiment and relevance to the SAMC phenotyping experiments is unclear and less accessible. Results appear slightly out of place within SAMC narrative. Although characterization of early infiltrating myeloid populations in dura is interesting, the majority of SAMCs appear to be of resident origin, likely forming from a pre-existing cell population. For example, osteopontin-positive SAMCs (one of the main functional markers identified) appear to all be nearly all resident derived.

Reply: We thank the reviewer for this comment and agree that our manuscript essentially has two parts: 1) a more descriptive part focussed on characterizing and identifying changes in leukocytes in parenchyma and border compartments induced by stroke using different techniques (Flow Cytometry, scRNA-Seq and Histology) and 2) a more discovery part that

uses the findings from the first part to describe SAMC and then subsequently dissect their phenotype and function in considerable detail. We believe that the Flow Cytometry part provides a reference to the field and supports the findings from the scRNA-Seq. experiments although mechanistic insights from it are - admittedly - limited. We have now extended the second part of the manuscript to provide many more details on the SAMC phenotype and the mechanism behind it.

8. No true functional assay of the SAMC, only characterization of their functionality based on IHC markers and expressional changes. Their importance is only hinted at through references and association, and never truly demonstrated to have direct impact on stroke pathology/outcome/etc.

Reply: We thank the reviewer for raising this important issue. To address this, we now provide evidence that the SAMC phenotype is associated with lipid-phagocytosing function in mice and humans (Suppl. Fig. 6). We first performed histological staining of SAMC-specific markers together with BODIPY or Perilipin-2, both of which specifically label lipid droplets (Suppl. Fig. 6). BODIPY⁺ lipid droplets were preferentially found in SAMC marker positive cells within the infarcted area (Suppl. Fig. 6A). Of note, Perilipin-2 was frequently expressed among cells with fluorescently labelled myelin indicating phagocytosis of myelin structures (Suppl. Fig. 7A). These findings were conserved in photothrombotic brainstem ischemia, in which the white matter is primarily affected (Suppl. Fig. 7B), and could be replicated in human tissue (Suppl. Fig. 6C). Of note, we found that blocking M-CSF by administration of a specific monoclonal antibody ameliorates structural and functional stroke outcome supporting the functional relevance of SAMC in stroke (Fig. 5). In detail, in mice treated with the SAMC marker specific antibody we found a trend towards a decreased amount of Perilipin-2⁺ cells (Fig. 5D,H) and in part of lipid droplets (Fig. 5D). Hence, SAMC may represent a potential future treatment target.

9. The term “Stroke associated myeloid cells” will prove to be controversial, as this cellular phenotype may only apply to this specific stroke model (animal, timeframe, methodology, etc). It remains to be seen if these SAMC signatures are truly held constant across all ischemic stroke models, and are relevant to human pathology. While the author’s showed that the presence of these SAMC markers exist in human brain stroke samples, without further studies and perhaps a true single-cell characterization of leukocytes across human stroke patients, we should be hesitant to broadly classify these as purely “stroke associated” as there is significant overlap with other degeneration associated phenotypes. This naming of phenotypical shifts in immune cell populations however is not uncommon in the field, but the name SAMC may prove to be too broad of a classification.

Reply: We thank the reviewer for this comment. First, we now confirmed the presence of the SAMC phenotype in other species and models. Of note, we replicated successfully the presence of SAMC-like myeloid lineage cells across different models of cerebral ischemia, mouse strains, and species (Suppl. Fig. 8). In detail, we stained SAMC markers in models of photothrombotic brainstem ischemia in rats, MCAO in rats and MCAO in mice with the different Balb/c genetic background. We confirmed the presence of myeloid lineage cells that express the SAMC-specific markers *Osteopontin/Spp1*, *LPL/Lpl*, *M-CSF/Csf1*, and *ADAM8/MS2* in all these models, strains, and species. We now depict these confirmatory data in Suppl. Figure 7 and emphasize it in the revised version of the manuscript as follows: ‘We next investigated the SAMC phenotype in other species and stroke models. First, we stained coronal brain sections of mice with the different Balb/C genetic background 24h after

30 min MCAO for Osteopontin/*Spp1*, LPL/*Lpl* and M-CSF/*Csf1* together with pan-myeloid F4/80 (Suppl. Fig. 7A). We found all of these markers being consistently expressed by F4/80⁺ cells within the infarction core. In addition, we induced photothrombotic brainstem ischemia in rats, and found the SAMC-markers Osteopontin/*Spp1*, M-CSF/*Csf1*, and ADAM8/*Adam8* frequently expressed by IBA1⁺ cells within the infarcted brainstem parenchyma (Suppl. Fig. 7B). After induction of MCAO in rats, the SAMC-markers Osteopontin/*Spp1*, M-CSF/*Csf1* and MMP12/*Mmp12* were also expressed across Iba-1-positive cells that were morphologically classified as microglia or macrophages (Suppl. Fig. 7C). In summary, we replicated the presence of SAMC-like myeloid lineage cells across different models of cerebral ischemia, mouse strains, and species. Second, we fully agree with the reviewer that the phenotype we denote as stroke-associated myeloid cells (SAMC) is likely not a distinct population, but rather a phenotype instructed onto the myeloid lineage by the ischemic CNS tissue. Accordingly, we found that these cells are derived from both resident microglia and peripheral monocytes/macrophages. We now emphasize this aspect in the text and discuss it in more detail in the discussion section on page 12 of the manuscript (see also comment #4 by reviewer #1) as follows: ‘Second, using single cell transcriptional profiling, we identify and subsequently confirm a novel phenotype which we name stroke-associated myeloid cells (SAMC) instructed onto the myeloid lineage by the ischemic CNS tissue. We find that these cells are not likely an ontogenetically homogeneous population, but are derived from both resident microglia and peripheral monocytes/macrophages. The core gene signature comprising *Spp1*, *Fabp5*, *Gpnmb*, *Ctsb*, *Ctsl*, *Lgals3*, *Lpl*, *Fth1*, *Cd63* and *Ctsd* distinctly characterized SAMC. We speculate that the ischemic tissue instructs this phenotype onto myeloid cells of different ontogenetic origin.’

10. 24hr timepoint for the scRNAseq tissue collection may be too early and narrow of timepoint to gain a complete accurate picture of SAMC phenotype shift. For example, the presence of SAMC marker positive cells in the parenchyma is much higher at 72hr across all markers investigated (Figure 4), indicating that later timepoints may be more important for the purposes of SAMC classification, not 24hr.

Reply: We thank the reviewer for this comment. We now provide scRNA-seq data at 72 hours in Figure 2. Notably, the SAMC cluster was absent in non-ischemic samples, but was induced at both time points after experimental stroke (24 and 72 hours). SAMC are thus highly stroke specific and are conserved during different post-ischemic time points. The largest increase of SAMC population was observed during the first 24 hours after stroke. Please also refer to point #5.

11. Figure 1B/C – Significance markers would be useful

Reply: We thank the reviewer for this comment. We have extended the flow cytometry data and depict them as scatter plots with significance markers in Figure 1 and Suppl. Figure 3. Please also refer to point #4.

12. Figure 1C – Color pallet makes interpretation difficult, consider different more easily differentiable line configurations.

Reply: We have adapted the color definition and line configurations accordingly.

13. Figure 1C – smaller immune cell populations (per cell count) are difficult to differentiate, consider alternative methods to display data. For example, supplemental figure 3B.

Reply: We have adapted the presentation of our data accordingly.

14. Page 5 (Line 13) – Misspelled stroke as: “Stoke”

Reply: We apologize for this mistake. We have corrected spelling accordingly.

15. P.2 line 5, p4: “deeply characterized” should be rephrased (performed in-depth characterization?)

Reply: We have adapted wording accordingly.

16. Abstract: SAMC should not be abbreviated here

Reply: We have adapted wording accordingly.

17. p12 should say that this is a transient MCAO model and they should include further details regarding following the STAIR criteria for stroke studies (number of mice seems to be low -4-6-, unbiased analysis, justification for males, etc)

Reply: We thank the reviewer for this important comment. First, we increased n-numbers of mice used in each experiment. Second, our experiments were performed according to the recent extra recommendations from STAIR XI regarding studies of basic mechanisms. In detail, the sample size for the chosen end point was determined by using power calculation. Furthermore, endpoints in our study (structural and functional outcome) reflect the mechanism under study (neuroprotection by phagocytosis). We now discuss the STAIR criteria and the limitations of all stroke models used for our experiments on page 13 of the manuscript as follows: ‘Rodent stroke models are often limited in understanding human stroke pathology. We aimed to address this issue by performing a power calculation and blinded assessments as recommended in the Stroke Academic Industry Roundtable (STAIR) criteria^{41,42}. A potential weakness of our study is that we chose young male mice in order to reduce variables with sex differences due to estrous cycle, and hence to reduce the number of animals used for ethical issues. Furthermore, the use of healthy animals without comorbidities may influence the structural and functional outcome after stroke. We aimed to mitigate this short-coming by confirming the presence of SAMC phenotype cells in several rodent models of stroke, strains, and across species.’. We also now provide data from several additional rodent models of stroke (photothrombotic brainstem ischemia in rat and MCAO in rat) and from MCAO in the Balb/c mouse strain in Suppl. Figure z. We confirmed the presence of SAMC phenotype cells in all these models, strains, and species. Please also refer to point #9.

Reviewer 2:

1. In this study Beuker and colleagues focus on a highly topical area, namely the immune cell landscape in barrier sites of the brain. This area has drawn increasing interest across multiple disease areas in recent years, aided by major advances in technology, including the ability to much better differentiate cell subtypes through single cell transcriptomics etc.

Reply: We thank the reviewer for the positive evaluation of our study.

2. The most noteworthy finding is the demonstration of a specific myeloid population within the barrier structures (pia) of the brain and in the CNS parenchyma that is largely derived from brain resident microglia. The authors define this population as stroke-associated myeloid cells (SAMC). Through detailed expression analyses it is further demonstrated that this cell population has a phagocytotic phenotype and the authors speculate this may be geared towards the clearing of myelin debris resulting from the death of neurones as a result of the stroke. However, no functional data is provided to substantiate this hypothesis, although immunofluorescence staining supports the concept. SAMC are also shown to express osteopontin, which the authors suggest could indicate a later pro-regenerative role for these cells, though this is very speculative and again not supported by functional data. The inclusion of data from post-mortem stroke brains to support the findings in the mouse model is noteworthy and strengthens the translational relevance of the work.

Reply: We thank the reviewer for this important comment. We now provide evidence that the SAMC phenotype is associated with lipid-phagocytosing function (Suppl. Fig. 6). We first performed histological staining of SAMC-specific markers together with BODIPY or Perilipin 2, both of which specifically label lipid droplets (Suppl. Fig. 6). BODIPY⁺ lipid droplets were preferentially found in SAMC marker positive cells within the infarcted area (Suppl. Fig. 6A). Of note, Perilipin-2 was frequently expressed among cells with fluorescently labelled myelin indicating phagocytosis of myelin structures (Suppl. Fig. 6A). These findings could be replicated in human tissue (Suppl. Fig. 6C). To validate the lipid-phagocytosing SAMC phenotype, we now provide data on in vitro lipid phagocytosis assays. Of note, the percentage of SAMC cells presenting diffused fluorescence indicating lipid clearance was increased in comparison to non-SAMC cells (Suppl. Fig. 6D). Most importantly, we find that blocking M-CSF by administration of a specific monoclonal antibody ameliorates structural and functional stroke outcome presumably by decreasing the amount of Perilipin-2⁺ cells and in part of lipid droplets (see Fig. 5). We thus provide data on the functionality of this phenotype and SAMC may represent a potential future treatment target. We further agree with the reviewer that osteopontin (encoded by SPP1) is a pleiotropic cytokine and is not sufficient to delineate the SAMC phenotype. We have therefore considerably down-phrased the importance of osteopontin in the manuscript and deleted this section from the discussion.

3. The conclusions are largely supported by the data, albeit the authors are speculative about potential functions of SAMC, without providing direct mechanistic evidence. Also, young male mice are used throughout. Stroke is largely an age-associated disease with co-morbidities. It is now well described that these co-morbidities can significantly influence the 'baseline' inflammatory status and subsequent response to stroke. This is not mentioned by the authors and should at least be acknowledged in the discussion.

Reply: We thank the reviewer for this important comment. We have amended the discussion section on pages 12 and 13 of the manuscript to acknowledge the limitations of rodent stroke models in understanding human stroke pathology as follows: "Rodent stroke models are often limited in understanding human stroke pathology. We aimed to address this issue by performing a power calculation and blinded assessments as recommended in the Stroke Academic Industry Roundtable (STAIR) criteria^{41,42}. A potential weakness of our study is that we chose young male mice in order to reduce variables with sex differences due to estrous cycle, and hence to reduce the number of animals used for ethical issues. Furthermore, the use of healthy animals without comorbidities may influence the structural and functional outcome after stroke. We aimed to mitigate this short-coming by confirming

the presence of SAMC phenotype cells in several rodent models of stroke, strains, and across species.”. Please also refer to point #9 and #17 by reviewer #1. The aspect of potential functional relevance of the SAMC phenotype is discussed in detail in point #8 by reviewer #1 and point #2 by reviewer #2.

4. In some experiments the sample size is low and the authors should provide details of how the number of animals used was determined.

Reply: We thank the reviewer for this comment. As also suggested in several comments by reviewer #1 we have considerably increased cell numbers in scRNA-seq, added further stroke models, and increased the number of biological replicates in all flow cytometry-based experiments.

REVIEWERS' COMMENTS

Reviewer #1 (Remarks to the Author):

My concerns have been addressed in the revision.

Reviewer #2 (Remarks to the Author):

A comprehensive response to all reviewer comments is provided. The revisions include the addition of new data and increased sample sizes for some studies, thus providing stronger evidence for these findings. The new data and additional text in the revised manuscript address all the issues raised on the original manuscript.

Reviewer Comments and Response

Reviewer 1:

1. My concerns have been addressed in the revision.

Reply: We thank the reviewer for the positive evaluation of our study.

Reviewer 2:

1. A comprehensive response to all reviewer comments is provided. The revisions include the addition of new data and increased sample sizes for some studies, thus providing stronger evidence for these findings. The new data and additional text in the revised manuscript address all the issues raised on the original manuscript.

Reply: We thank the reviewer for the positive evaluation of the revised version of the manuscript.